# Towards Large-Scale Daily Snow Density Mapping with Spatiotemporally Aware Model and Multi-Source Data

Huadong Wang[1], Xueliang Zhang[1], Pengfeng Xiao[1,2], Tao Che[3], Zhaojun Zheng[4], Liyun Dai[3], Wenbo Luan[1]

[1]Jiangsu Provincial Key Laboratory of Geographic Information Science and Technology, Key Laboratory for Land Satellite Remote Sensing Applications of Ministry of Natural Resources, School of Geography and Ocean Science, Nanjing University, Nanjing, Jiangsu 210023, China.
[2]Jiangsu Center for Collaborative Innovation in Geographical Information Resource Development and Application, Nanjing, Jiangsu 210023, China
[3]Key Laboratory of Remote Sensing of Gansu Province, Heihe Remote Sensing Experimental Research Station, Cold and Arid Regions Environmental and Engineering Research Institute, Chinese Academy of Sciences, Lanzhou 730000, China.
[4]The National Meteorological Satellite Meteorological Center, Beijing 100081, China.

*Correspondence to:* Xueliang Zhang (zxl@nju.edu.cn)

**Abstract.** Snow density plays a critical role in estimating water resources and predicting natural disasters such as floods, avalanches, and snowstorms. However, gridded products for snow density are lacking for understanding its spatiotemporal patterns. In this study, considering the strong spatiotemporal heterogeneity of snow density, as well as the weak and nonlinear relationship between snow density and the meteorological, topographic, vegetation, and snow variables, the geographically and temporally weighted neural network (GTWNN) model is constructed for estimating daily snow density in China from 2013 to 2020, with the support of satellite, ground, and reanalysis data. The leaf area index of high vegetation, total precipitation, snow depth, and topographic variables are found to be closely related to snow density among the 20 potentially influencing variables. The 10-fold cross-validation results show that the GTWNN model achieves an $R^2$ of 0.531 and RMSE of 0.043 g/cm$^3$, outperforming the geographically and temporally weighted regression model ($R^2$ = 0.271), geographically weighted neural network model ($R^2$ = 0.124), and reanalysis snow density product ($R^2$ = 0.095), which demonstrates the superiority of the GTWNN model in capturing the spatiotemporal heterogeneity of snow density and the nonlinear relationship to the influencing variables. The performance of the GTWNN model is closely related to the state and amount of snow, in which more stable and plentiful snow would result in higher snow density estimation accuracy. With the benefit of the daily snow density map, we are able to obtain knowledge of the spatiotemporal pattern and heterogeneity of snow density in China. The proposed GTWNN model holds the potential for large-scale daily snow density mapping, which will be beneficial for snow parameter estimation and water resource management.

## 1 Introduction

Seasonal snow cover occupies an important position in the global surface energy balance, hydrological cycle, and climate system (Bormann et al., 2018; Hall and Qu, 2006; Hernández et al., 2015; Li et al., 2018), which accounts for approximately 50% of the land area in winter in the Northern Hemisphere (Frei and Robinson, 1999). Snowmelt water not only provides fresh water for one-sixth of the world's population, but also affects agriculture and ecosystems downstream (Barnett et al., 2005). Snow density is an important variable of snowpack, which influences the thermal, mechanical, and optical properties of snow layers (Surendar et al., 2015). It plays an important role in predicting natural disasters such as floods, avalanches, and snowstorms, establishing hydrological models, as well as water resources management (Fayad et al., 2017; Judson and Doesken, 2000; Roebber et al., 2003; Schweizer et al., 2003).

Snow water equivalent (SWE) is critical for evaluating the contribution of snow cover to water resources (Niedzielski et al., 2019; Varade et al., 2020), which can be obtained by multiplying snow depth (SD) and snow density. Since the 1970s, algorithms for retrieving SD using passive microwave remote sensing have been continuously optimized (Chang et al., 1987;

Gharaei-Manesh et al., 2016; Pulliainen et al., 1999). However, the SWE products obtained by passive microwave remote sensing are mostly produced with a fixed value of snow density (Che et al., 2016; Pulliainen et al., 2020), due to the limited knowledge of snow density distribution. The snow density changes with time, and there is also strong spatial heterogeneity (Yang et al., 2019; Zhong et al., 2014). Snow density serves as a key variable for the accuracy of SWE products (McCreight and Small, 2014; Zaremehrjardy et al., 2021). For example, Yang et al. (2020) evaluated the accuracy of the GlobSnow-2 SWE product and found that using a fixed snow density would result in overestimated SWE in China. Therefore, the daily gridded snow density product will benefit for the estimation of SWE.

The fresh snow density is determined by the environment of falling snow, such as air temperature, relative humidity, and air pressure. After that, the size, shape, and packing of snow crystals are affected by the accumulation, sublimation, and melting of snow crystals on the surface, which leads to changes in snow density (Nakaya, 1951; Roebber et al., 2003). Snow density will also increase with snow age and SD due to the metamorphism and compaction, and the change rate is mainly influenced by melt-refreeze events and wind erosion (Bormann et al., 2013; Meløysund et al., 2007). For example, the snow density in Northwest and Northeast China from 1999 to 2008 was found to be closely related to SD, as indicated by stepwise regression analysis of snow density and temperature, precipitation, SD, and wind speed (Dai and Che, 2011).

The terrain and surface types also play an important role in snow density (Clark et al., 2011; Judson and Doesken, 2000). For example, snow density of tundra snow was found to be lower at higher elevations, and even decreased by approximately 0.006 g/cm$^3$ with each 100 m increase in elevation in the former Union of Soviet Socialist Republics (USSR) (Zhong et al., 2014), which is indirectly affected by energy balance, temperature decreases with elevation in general (Elder et al., 1998). The indirect effect of slope on snow density includes two ways, one is redistribution of snow via avalanching and wind transport, and another is the amount of radiation received, which results in changes in snow grain size, porosity, and density. Slopes with high radiation inputs will be more likely to have snow melt, introducing liquid water into the snow, which also increase snow density by filling the pore space with liquid water (Wetlaufer et al., 2016). The average snow density in forest areas was 8%–13% less than that in open areas (Zhong et al., 2014), and these observed density differences are attributed to either mass, delivery, wind, or radiation effects (Bonner et al., 2022). Mass effect is a reduction in the snow mass due to canopy interception loss, with lower compaction rates and snow density. Delivery effect refers to that snow is trapped by the canopy and then delivered to the underlying snowpack, either as unloaded snow or draining melt water. Wind effect occurs when wind speed is reduced by forest obstruction, resulting in a higher snow density relative to open areas because of wind packing. Radiation effect can control snow layer temperature, and melt-refreeze cycles to change snow density (Essery et al., 2008; Storck et al., 2002; Winstral and Marks, 2014).

The spatial and temporal differences in the distribution of multiple complex influencing variables result in obvious spatiotemporal heterogeneity of snow density. In the former USSR, snow density will increase with latitude, while the snow density of the Altai Mountains in China is more related with longitude (Zhong et al., 2014, 2021b). The average snow density shows obvious inter-monthly variation in the three major seasonal snow cover areas of China from 1957 to 2009 (Ma and Qin, 2012), and the monthly maximum snow density moved from north to south from October to January (Dai and Che, 2011). Moreover, this spatiotemporal heterogeneity is also reflected in the relationship between snow density and its influencing factors. The SD is often used to estimate snow density by different models (Lundberg et al., 2006; Sturm et al., 2010). However, the relationship between them is not robust in different time and space, where the positive and negative relationship and the significance of correlation coefficient vary greatly at small scales (López-Moreno et al., 2013).

To understand the spatiotemporal heterogeneity of snow density, people often use the ground observation data, but it is difficult to achieve large-scale monitoring due to the complex environment and limited number of the stations. One method to explain the spatial and temporal variations in snow density is to use a physical model, such as the coupled energy and mass-balance model ISNOBAL (Hedrick et al., 2018; Marks et al., 1999), which can explicitly simulate a number of snowpack properties including snow density and SWE at the regional scale, and add a physical basis of energy exchange in the snowpack.

However, snow density physical models are complex and cannot achieve large-scale spatialization of snow density (Raleigh and Small, 2017). Another common method is to use statistical models trained by the climatic and snow variables to produce snow density map, such as multiple linear regression (MLR) and binary regression tree analysis (Meløysund et al., 2007; Mizukami and Perica, 2008; Wetlaufer et al., 2016). However, the simple statistical models may not well capture the complicated nonlinear relationship of multiple influencing variables for snow density. More importantly, the models were

mostly constructed for each observation independently and neglected the spatiotemporal heterogeneity of snow density as well as the relationship to its influencing variables.

Geographically weighted regression (GWR) is a model that considers spatial heterogeneity by using local multiple linear regression technology (Fotheringham et al., 1998). To further incorporate temporal dependency, geographically and temporally weighted regression (GTWR) model has been introduced for many disciplines, such as meteorology, hydrology, and social

economics (Chen et al., 2017; He and Huang, 2018; Huang et al., 2010). The machine learning approaches such as Random Forests (RF) (Breiman, 2001) and General Regression Neural Network (GRNN) (Specht, 1991) have become popular to fit nonlinear relationships, and it is in the initial stage for estimating snow density (Broxton et al., 2019). We can incorporate geographical and temporal weights into a neural network model to capture the spatiotemporally variable and nonlinear relationship between snow density and its influencing variables. In addition, considering the impact of different influencing

variables, the satellite data can provide information on the snow-related and topography-related variables, and the reanalysis data can provide information of the meteorology-related variables for estimating snow density based on the true value provided by ground observations. Consequently, to achieve large-scale snow density mapping, we can develop a geographically and temporally weighted neural network (GTWNN) model by considering the multiple influencing variables with the support of satellite, ground, and reanalysis data, which not only considers the spatiotemporal heterogeneity for snow density, but also

explains the nonlinear relationship between snow density and different influencing variables.

The main objectives of this study are (1) to develop a GTWNN model for improving snow density mapping by addressing the spatiotemporal heterogeneity and capturing the nonlinear relationship between snow density and its influencing variables; (2) to validate the effectiveness of the proposed model in various situations and to understand the relationship between snow density and its influencing variables; and (3) to achieve daily snow density mapping by integrating satellite, ground, and

reanalysis data and to understand the spatiotemporal pattern of snow density in China.

## 2 Study Area and Data

### 2.1 In Situ Snow Density

We aim to achieve snow density mapping in China, where Xinjiang, Northeast China-Inner Mongolia, and Tibetan Plateau are the three major regions with stable seasonal snow cover, covering a total area of approximately 4,200,000 km² (Huang et al.,

2016). Snow cover in other areas of China melts rapidly because of the relatively high temperature and is thus not viewed as stable seasonal snow cover. The daily SD and snow pressure measurements are collected from the China Meteorological Administration (CMA) to calculate snow density, including 984 stations from 2013 to 2020 (Figure 1). In addition, 585 snow pits of the snow survey dataset from measurement routes in typical regions from 2017 to 2019 are also used (Che, 2021), which are collected from the National Cryosphere Desert Data Center (http://www.ncdc.ac.cn). The ground observations are

concentrated in the snow season, with few observations in summer from June to August. Therefore, the study focuses on estimating snow density in the snow season from September to May of the next year. To further analyze the estimation results, the snow season is roughly divided into the snow accumulation period (September–November, autumn), the snow stable period (December–February of the next year, winter), and the snowmelt period (March–May, spring) according to the division of season (Ke et al., 2016).

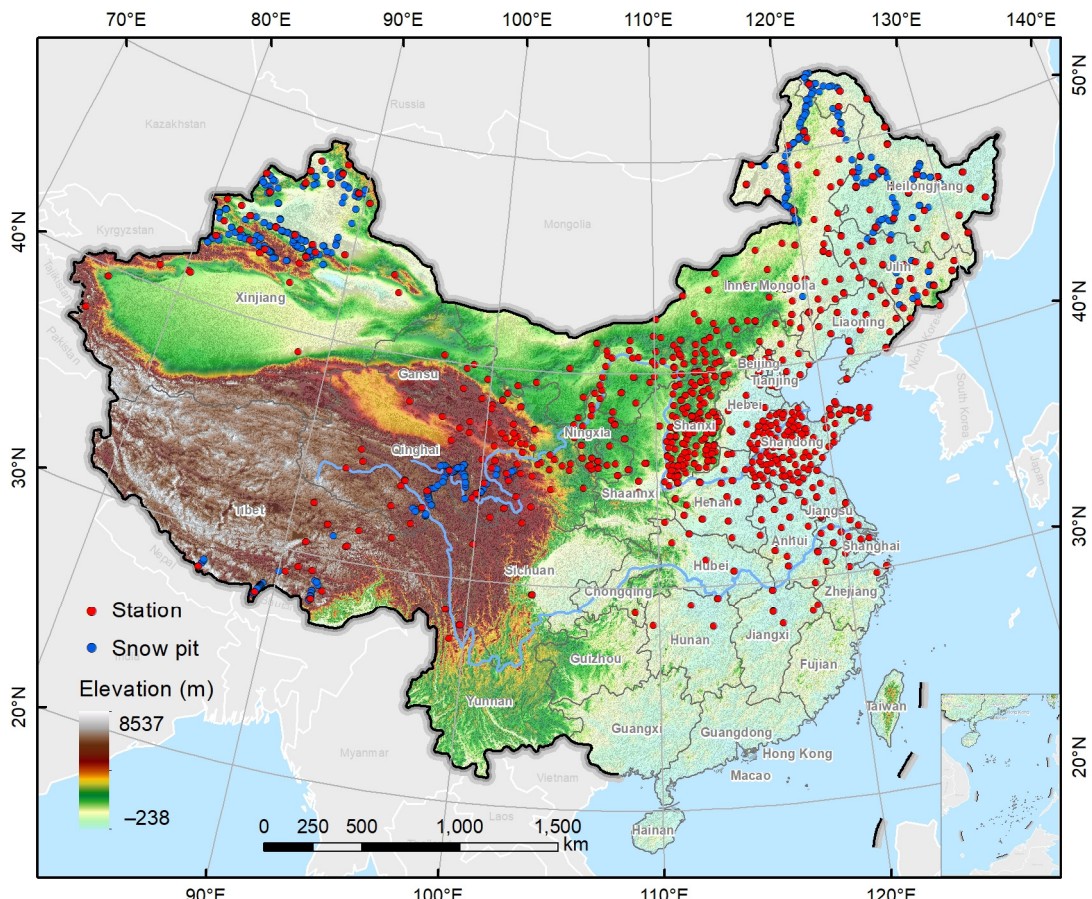


**Figure 1. Spatial distribution of collected ground observations of snow density.**

## 2.2 Satellite and Reanalysis Data

The ECMWF ERA-5 land hourly dataset is adopted to provide data on meteorological variables, vegetation variables, and some snow variables, which is a climate reanalysis dataset providing a consistent view of the evolution of land variables over

several decades at a spatial resolution of 0.1°×0.1° (https://cds.climate.copernicus.eu) (Muñoz-Sabater, 2019). We extract the 10 m u-component of wind (U10), 10 m v-component of wind (V10), 2 m temperature (T2M), surface pressure (SP), total precipitation (TP), snowmelt (SMLT), snowfall (SF), temperature of snow layer (TSN), snow evaporation (ES), and leaf area index of high vegetation (LAI_HV), and calculate their daily average or accumulation accordingly.

The satellite products of snow albedo (SA), snow depth (SD), and snow cover area (SCA) from 2013 to 2020 are collected

from the National Cryosphere Desert Data Center (http://www.ncdc.ac.cn) (Hao et al., 2021b; Xiao et al., 2020; Yang et al., 2019), with spatial resolutions of 0.01°, 25 km, and 5 km, respectively. Based on the SCA data, the snow cover duration (SCD) is calculated to account for the impact of snow duration on snow density. In addition, the MODIS land vegetation cover classification product (MCD12Q1) is used for obtaining the surface types, with the spatial resolution of 500 m.

The topographical variables of elevation are obtained from the Shuttle Radar Topography Mission (SRTM) digital

elevation model with a spatial resolution of 30 m, and then slope and aspect are derived based on the elevation.

## 2.3 Data Integration

Three kinds of data are used, including ground observation data, satellite data, and reanalysis data, where the ground observation data is used to provide the true value of snow density, and the satellite and reanalysis data are used to provide information of different influencing variables of snow density. Before the model development, data pre-processing is

conducted. Firstly, since the spatial resolution varies among the different influencing variables on snow density, they are resampled to 25 km for snow density mapping using average or accumulation resampling methods depending on the data type.

The spatial resolution of 25 km is determined to match that of most SD and SWE products by passive microwave remote sensing. The elevation and slope are resampled to 25 km by average, and the standard deviation of elevation (ELEVATION_STD) and slope (SLOPE_STD) are also calculated to reflect the topographic relief within the range of 25 km.

Accordingly, the ground observations of snow density measured at multiple sites are averaged for each 25 km grid cell. In addition, to eliminate the influence of different dimensions, the min-max normalization method is applied to normalize different influencing variables except for MCD12Q1 data. After that, we collect 16935 samples for model establishment and validation, where a sample refers to a grid cell with ground observations of snow density and its influencing variables.

## 3 Methodology

### 3.1 GTWNN Model

The GTWNN model is a spatiotemporally aware model composed of a geographically and temporally weighted (GTW) model to capture spatiotemporal heterogeneity and a generalized regression neural network (GRNN) to deal with the weak and nonlinear relationships between snow density and its influencing variables, including the meteorological variables, topographical variables, vegetation variables, and snow variables, which could be expressed as shown in Eq. (1), and its

schematic is shown in Figure 2.

$$snow\ density = f_{(S,T)}(x, y) \,, \tag{1}$$

where $snow\ density$ is the estimated snow density in each cell; $(S, T)$ presents the spatial and temporal distance between the sample point and the prediction point; $x$ refers to the influencing variables of snow density; and $y$ refers to the ground observation data.

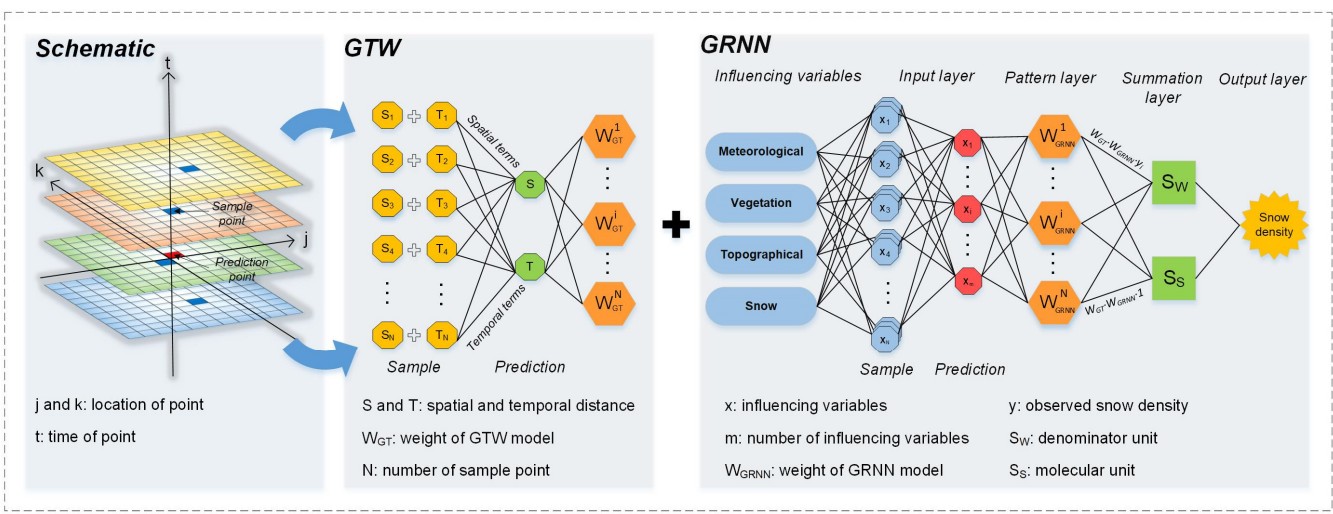


**Figure 2. Schematic of the GTWNN model for the estimation of snow density, where GTW refers to the geographically and temporally weighted, and GRNN refers to the generalized regression neural network.**

The GTW model explores the spatiotemporal heterogeneity through local weighting, which can assess the impact of sample points on prediction points in terms of the spatial and temporal distances (Figure 2). The weight of each sample point

is calculated by the commonly used bi-square function (Guo et al., 2008), as shown in Eq. (2) and (3).

$$d_{ST}^i = \sqrt{\left[\left(j_p - j_s^i\right)^2 + \left(k_p - k_s^i\right)^2\right] + \varphi(t_p - t_s^i)^2} \,, \tag{2}$$

$$W_{GT}^i = \begin{cases} [1 - \left(\frac{d_{ST}^i}{h_{ST}}\right)^2]^2, & d_{ST}^i < h_{ST} \\ 0, & d_{ST}^i \geq h_{ST} \end{cases} \,, \tag{3}$$

where $d_{ST}^i$ denotes the spatial and temporal distance between the $i_{th}$ ($i = 1, 2, …, N$) sample point ($s$) and the prediction point ($p$), in which $j$ and $k$ represent the location of point; and $t$ represents the time, as shown in Figure 2. $W_{GT}^i$ indicates the weight of the sample point in the GTW model. The spatiotemporal nonnegative parameters known as bandwidth $h_{ST}$ and scale factor $\varphi$ are the two key parameters in the GTW model (Huang et al., 2010). In essence, there are two weighting regimes for setting $h_{ST}$: fixed kernel and adaptive kernel. To reduce the "weak data" problem, that is, the spatial distribution of snow density observations is uneven, the adaptive kernel is selected to adapt the localization patterns of the observations by changing the kernel size automatically. The kernel will be large in regions with sparsely distributed observations and small when the data are abundant. The scale factor $\varphi$ balances the different effects of the spatial and temporal distances in their respective metric systems. When $\varphi = 0$, the temporal distance has no effect on the weight, indicating that the sample at any time will be considered. When $\varphi = \infty$, only the samples with the same date of the prediction point have an influence on the prediction, and GTW degrades into the classic geographically weighted (GW) model.

A common GRNN architecture consists of four layers (Li et al., 2020), as shown in Figure 2. The first layer is the input layer receiving the influencing parameters $x$, the number of neurons is equal to the input vector dimension $N$, and the number of influencing variables is $m$. The pattern layer is the radial base layer, and the weight of each neuron $i$ in the pattern layer $W_{GRNN}^i$ is calculated by the difference in the influencing variables between the sample and prediction points using a Gaussian function:

$$W_{GRNN}^i = e^{-\frac{1}{2}\left(\frac{D(x_p - x_s^i)}{spread}\right)^2},$$  (4)

where $x_s^i$ and $x_p$ represent the influencing variable values of the sample and prediction points; $D()$ refers to the Euclidean distance; and $spread$ is a parameter to control the smoothness of the fitting function. Successively, there are two kinds of neurons in the summation layer. One is the denominator unit ($S_W$) for calculating the algebraic sum of each neuron, in which the weight of each neuron $i$ in the summation layer is the snow density of sample point $y_i$, and the other is the molecular unit ($S_S$) for calculating the weighted sum of the pattern layer neurons. Finally, by combining the GTW and GRNN, the output snow density of the prediction point can be expressed as Eq. (5), where $W_{GT}^i$ captures the spatiotemporal heterogeneity and $W_{GRNN}^i$ relates to the nonlinear relationship between snow density and its influencing variables.

$$snow\ density = \frac{S_W}{S_S} = \frac{\sum_i^N (W_{GT}^i \cdot W_{GRNN}^i \cdot y_i)}{\sum_i^N (W_{GT}^i \cdot W_{GRNN}^i \cdot 1)},$$  (5)

**3.2 Parameter Determination and Model Evaluation Method**

There are three essential parameters in GTWNN model, including the spatiotemporal bandwidth $h_{ST}$ and the scale factor $\varphi$ of the GTW model, and the $spread$ of GRNN model. To evaluate the model performance as well as to determine the optimal parameters, the 10-fold cross-validation technique is adopted (Fotheringham et al., 2003; Rodriguez et al., 2010); that is, all the collected samples are randomly divided into 10 folds, nine folds are exploited for the model fitting, and one fold is used for the validation. The above steps are repeated 10 times so as to evaluate the model performance on each fold of the validation samples. Finally, a scale factor $\varphi$ of 0.01, $spread$ of 0.5, and an adaptive bandwidth regime $h_{ST}$ of 8 are obtained, which can achieve the best performance.

In addition, the coefficient of determination ($R^2$, unitless), the mean absolute prediction error (MAE, g/cm$^3$), and the root mean squared prediction error (RMSE, g/cm$^3$) are adopted to evaluate the performance of the GTWNN model.

## 4 Results

### 4.1 Descriptive Statistics of Ground Observations

Snow density has strong spatiotemporal heterogeneity, and we calculated statistics of the 16935 samples generated from ground observations in terms of the snow density and the number of observations in different years, months, and snow cover regions, as shown in Figure 3, which show the dispersion and variation fluctuations in snow density and can be used to verify the results of snow density mapping.

The snow density averaged in China from 2013 to 2020 is 0.140 g/cm³. The mean and median snow density values change 215 slightly from 2013 to 2020 except for the fluctuation in 2019 with a snow density of 0.180 g/cm³. For the monthly variation, the mean snow density tends to increase from 0.120 g/cm³ in October with the accumulation of snow and achieves the highest value of 0.162 g/cm³ in March. Snow density also varies spatially. Among the three major snow cover regions, Xinjiang has the largest mean snow density (0.159 g/cm³), successively followed by Tibetan Plateau and Northeast China-Inner Mongolia.

In addition to the spatiotemporal variation in snow density, the number of ground observations also varies. The number 220 of observation samples from 2013 to 2018 ranged from 2250 to 3250 but decreased to less than 750 in 2019 and 2020, mainly because of the lack of observations at many meteorological stations. The number of observation samples varies in different months mainly because of the richness of snow, which is higher in the snow stable period than in other periods. The number of observation samples varies spatially mainly because of the distribution of meteorological stations, where Northeast China-Inner Mongolia has the most stations, followed by Xinjiang and Tibetan Plateau.

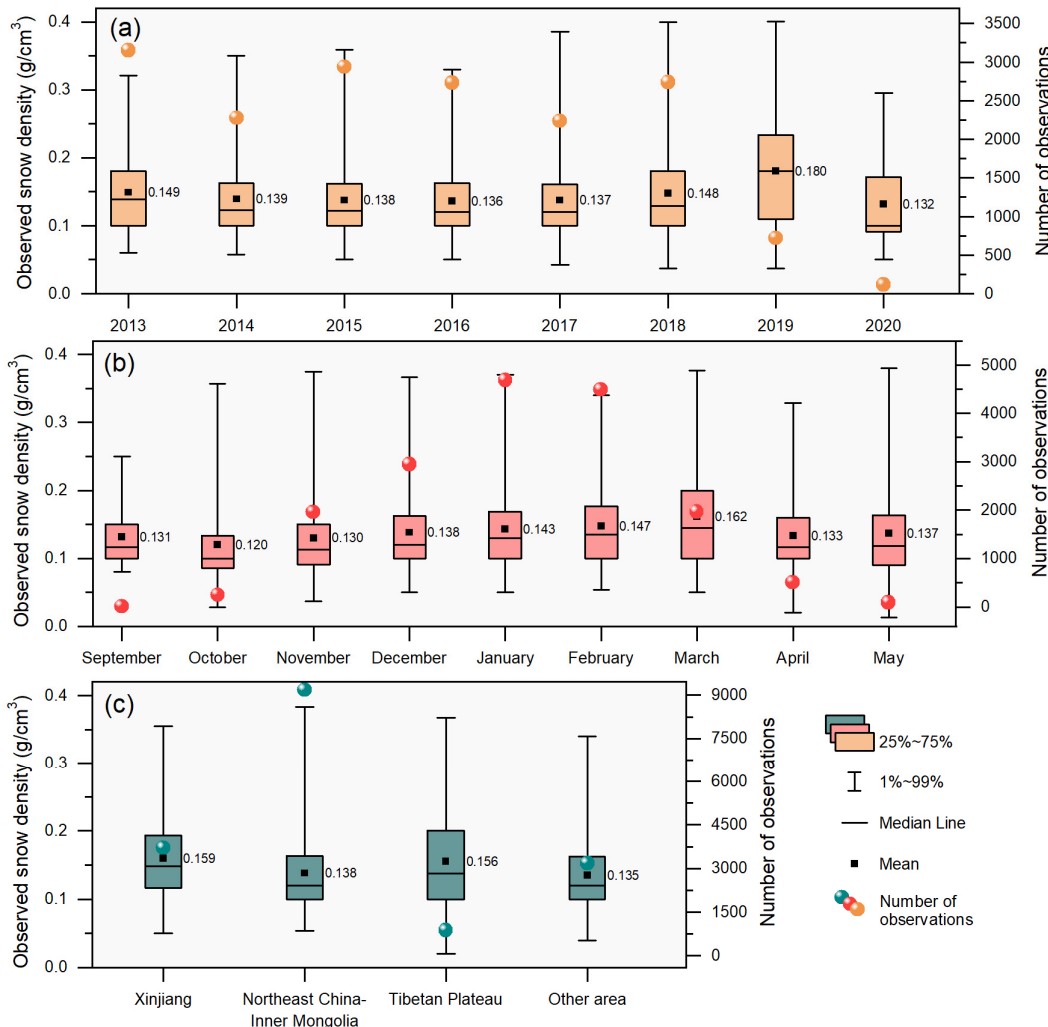


**Figure 3. Descriptive statistics of the snow density and the number of ground observations in different years (a), months (b), and snow cover regions (c).**

## 4.2 Model Validation

### 4.2.1 Relationship between Snow Density and Its Influencing Variables

The Pearson correlation coefficient between snow density and its influencing variables is calculated to indicate the importance of the variables in each month, as shown in Figure 4a, where September and May are not included because of the small number of ground observations. The influencing variables and the corresponding correlation coefficient values are various in different months because of the heterogeneity of snow. In addition, we calculate the average value from October to April for the positive and negative correlation coefficients, respectively, to indicate the importance of each influencing variable

for snow density. We also count the number of months with positive or negative correlations and mark the correlations that appear in more months as "main correlation", to clearly show the relationship between snow density and different influencing variables, as shown in Figure 4b. In general, the correlations between snow density and all influencing variables are very weak, with the maximum average correlation coefficient of only 0.123, which indicates the great difficulty for the estimation task of snow density.

For the 8 snow variables, SD shows apparently higher importance because it has the larger average correlation coefficient of 0.087, followed by ES and SMLT with average correlation coefficient of 0.082. It is noted that snow density is mainly negatively correlated with SF, SA, and SCD, and positively correlated with other snow variables, indicating that the less new snowfall, more snowmelt, and deeper snow depth tend to have higher snow density. Among the 5 meteorological variables, TP has the highest average correlation coefficient of 0.110, indicating that higher precipitation can increase snow density. All

five topographical variables show high positive correlation, with average correlation coefficient value of approximately 0.1. Surprisingly, the variable LAI_HV has the largest positive correlation coefficient among all the variables, indicating the importance of vegetation for snow density estimation. In summary, LAI_HV has the strongest correlation with snow density, followed by the TP, SD, and topographic variables among the 20 variables.

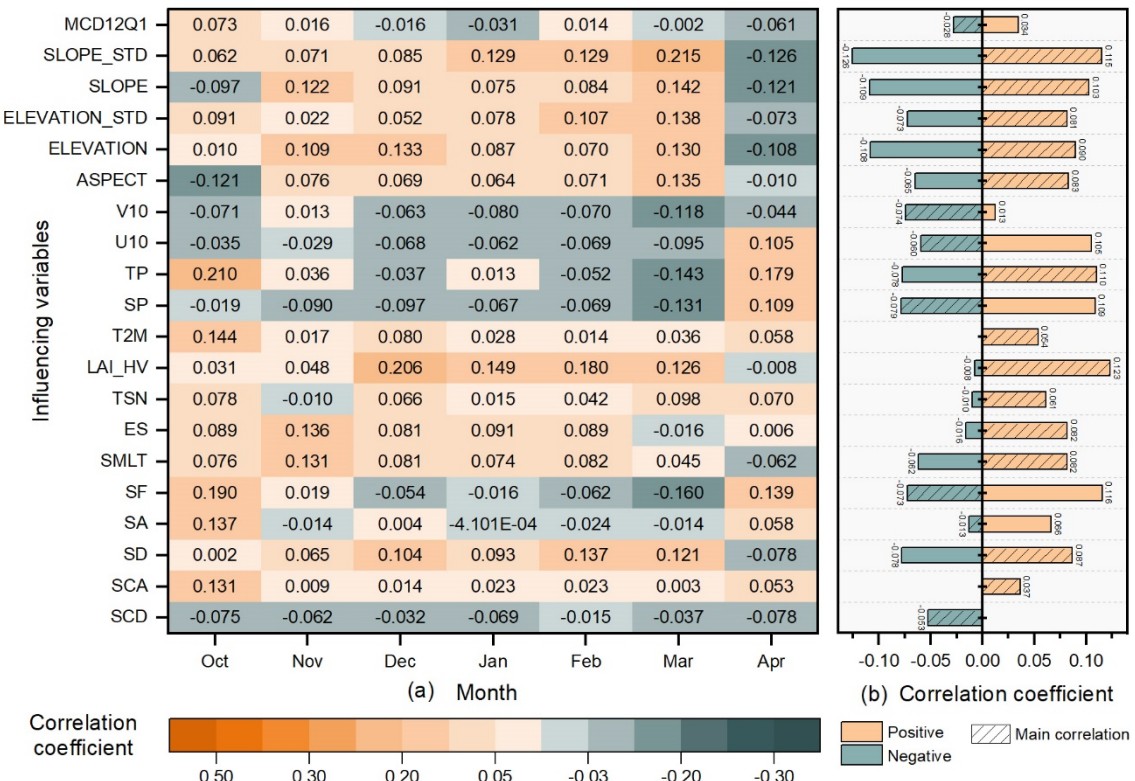

**Figure 4. Correlation coefficients between snow density and its influencing variables in each month (a), and the average value of the positive and negative correlation coefficients, where the main correlation marked as shade refers to the positive or the negative correlation that occurs in more months than the other (b).**

### 4.2.2 Accuracies of GTWNN Model in Different Regions

The snow density estimation accuracies of the GTWNN model are assessed over China (Figure 5a) and different snow cover regions, including Xinjiang (Figure 5c), Tibetan Plateau (Figure 5d), Northeast China-Inner Mongolia (Figure 5e), and other areas with instantaneous snow cover (Figure 5f). To clearly present the snow density errors between estimated values and observed values, the frequency of errors together with the Gaussian fitting curve are calculated and shown in Figure 5b.

Figure 5a shows that the $R^2$, MAE, and RMSE values over China are 0.531, 0.028 g/cm$^3$, and 0.043 g/cm$^3$, respectively, which indicates that the GTWNN model is able to account for 53.1% of daily snow density variations. The linear fitting curve is very close to the 1:1 line with a slope of 0.906, and most of the points are concentrated on the trend line, especially in the range of 0.1–0.25 g/cm$^3$. Figure 5b further demonstrates the concentration since most of the errors are smaller than ±0.04 g/cm$^3$. It is noted that the number of estimated values greater than 0.3 g/cm$^3$ is rare, indicating that the GTWNN model would underestimate the very large snow density.

Among the three stable seasonal snow cover regions, Xinjiang achieves the highest $R^2$ of 0.633 and the lowest RMSE of 0.038 g/cm$^3$ and MAE of 0.022 g/cm$^3$, followed by Northeast China-Inner Mongolia. Although Northeast China-Inner Mongolia has more observation samples than Xinjiang, the lower accuracies in Northeast China-Inner Mongolia would mainly be caused by greater forest and less stable snow cover in Northeast China-Inner Mongolia than in Xinjiang. The Tibetan Plateau has the lowest $R^2$ of 0.517 and the highest RMSE and MAE, which is mainly caused by the high variation fluctuations of snow density and sparse meteorological stations, as indicated in Figure 3c. Compared with the stable regions, the estimation accuracies in other areas of China are apparently lower, with an $R^2$ of 0.183, which is caused by the rapid melting of snow and sparse observations.

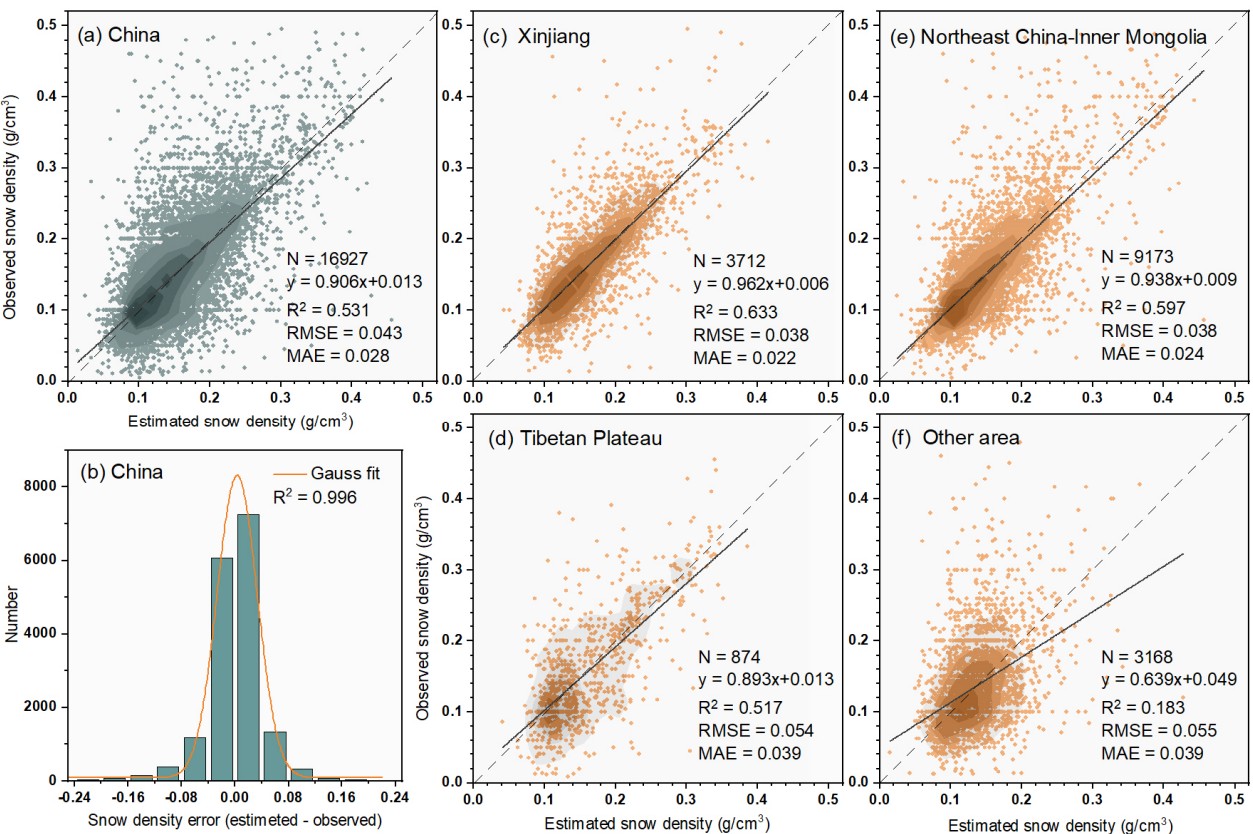

**Figure 5. Accuracies of the estimated snow density in China (a) and different snow cover regions (c, d, e, and f) and the histogram of the snow density errors between estimated and observed values (b).**

### 4.2.3 Accuracies of GTWNN Model in Different Months

The snow density estimation accuracies of each month are assessed over the entire study area to reveal the effectiveness of GTWNN model in different months, as shown in Table 1. In the snow season, the snow stable period achieves the best estimation performance with $R^2$ 0.587, RMSE 0.038 g/cm$^3$, and MAE 0.023 g/cm$^3$. Within the snow stable period, the highest $R^2$ (0.597) appears in January, together with the lowest RMSE (0.037 g/cm$^3$) and MAE (0.022 g/cm$^3$), which is because of the stable snow state and the relatively sufficient observation samples.

The accuracies in the snow accumulation and snowmelt period are inferior to those in snow stable period, which is mainly caused by the relatively rapid changes in snow as well as the sparser observations, especially for the months of October, April, and May. It is noted that the observation error cannot be ignored, which may be caused by the less snow in the early stage of snow accumulation period, or the large water content when the snowmelt period, making the observation more difficult. The accuracies in November and March are apparently higher than those in October, April, and May, mainly because the snow in these two months does not change so fast. The accuracies in September are not involved in the analysis because there are only 11 observations.

According to above results, we can safely conclude that the snow density estimation achieves the best performance in the snow stable period over the entire study area, and the estimations in November and March are also acceptable considering both the accuracies and the number of observations.

**Table 1. Accuracies of snow density estimation in different months.**

| Snow cover period | Month | Number | Slope | $R^2$ | RMSE | MAE |
|---|---|---|---|---|---|---|
| Snow accumulation period | September | 11 | 0.141 | 0.005 | 0.051 | 0.038 |
| | October | 253 | 0.559 | 0.120 | 0.058 | 0.041 |
| | November | 1960 | 0.862 | 0.475 | 0.048 | 0.032 |
| Snow stable period | December | 2952 | 0.939 | 0.584 | 0.040 | 0.025 |
| | January | 4688 | 0.940 | 0.597 | 0.037 | 0.022 |
| | February | 4491 | 0.916 | 0.577 | 0.038 | 0.024 |
| Snowmelt period | March | 1970 | 0.886 | 0.473 | 0.054 | 0.035 |
| | April | 510 | 0.362 | 0.056 | 0.064 | 0.046 |
| | May | 92 | 0.561 | 0.102 | 0.071 | 0.050 |

Furthermore, to clearly reveal the monthly accuracies of GTWNN model in different snow cover regions, the snow density estimation accuracies for each snow period and part of the months are assessed in different snow cover regions, as shown in Figure 6.

In different snow periods, the snow stable period also achieves the highest $R^2$ in different snow cover regions except for the Tibetan Plateau, as shown in Figure 6a. The snow in Xinjiang, Northeast China-Inner Mongolia, and other areas of China is concentrated in the coldest months, and the state of snow is more stable in these months, resulting in a higher $R^2$. The accuracies on the Tibetan Plateau decrease from the snow accumulation period to the snowmelt period. This is because snow accumulates early and disappears late in the Tibetan Plateau, and the shallow snow depths make it difficult to maintain snow cover, resulting in more snow in autumn and spring and less snow in winter (Li and Mi, 1983; Zhong et al., 2021a). In addition, the snow has large water content and changes rapidly in the snowmelt period, which results in a lower estimation accuracy in spring. Hence, combining the relatively large amount and the stable state of snow in the snow accumulation period, the snow density estimation accuracy is the highest in this period on the Tibetan Plateau.

We choose the three months of the snow stable period and November and March to analyze monthly accuracies in different regions because of the relatively higher overall accuracies in these months, as shown in Table 1, and the results are

shown in Figure 6b. In most cases, the accuracies in Xinjiang, Northeast China-Inner Mongolia, and other areas of China first increase and then decrease from November to March and achieve the highest accuracies in January or February, when the snow cover is plentiful and stable. However, the accuracy on the Tibetan Plateau changes oppositely and achieves the highest accuracy in November because of the specialty of the snow amount changes within a year, as discussed above.

Therefore, we conclude that the accuracies of the GTWNN model are generally related to the stability and the amount of snow. The snow density estimations achieve the highest $R^2$ in the snow stable period in Xinjiang, Northeast China-Inner Mongolia, and other areas of China because of the concentration and stability of snow in this period, and achieve the highest $R^2$ during the snow accumulation period in the Tibetan Plateau because of the relatively large amount and stability of snow in this period.

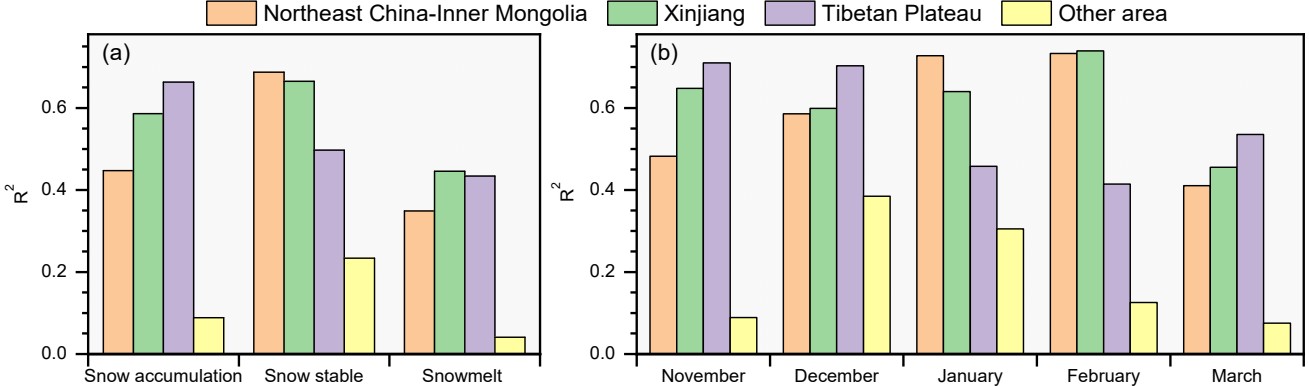

**Figure 6. Accuracies of snow density estimation in different snow periods (a) and months (b) over different snow cover regions.**

## 4.3 Model Comparison

### 4.3.1 Comparison with Other Regression Models

The GTWNN model is compared with five other regression models to demonstrate its advantages for snow density estimation
by capturing the spatiotemporal heterogeneity of snow density and its nonlinear relationship to influencing variables, as shown in Table 2. The models involved for comparison include the multiple linear regression (R) model, geographically weighted regression (GWR) model, geographically and temporally weighted regression (GTWR) model, general regression neural network (GRNN) model, and geographically weighted neural network (GWNN) model. It is noted that the original R and GRNN models are global regression models established on all samples, regardless of the geographical and temporal weights.
The R model captures the linear relationship between snow density and its influencing variables, and the GRNN has nonlinear mapping ability (Specht, 1991). Meanwhile, the GWR (Fotheringham et al., 1998) and GWNN models are spatial local models constructed from R and GRNN by setting a bandwidth $h$, in which the sample points have different weights ($W_G$) according to the spatial distance ($d_s$). The GTWR and GTWNN models further incorporate temporal dependencies, which adds a new scale factor $\varphi$ to balance the different weights of the spatial and temporal distances. The optimal parameters of the compared
models are determined by the 10-fold cross validation strategy as that for the GTWNN model.

The accuracies achieved by GTWNN are apparently higher than those achieved by GRNN and GWNN, which demonstrates the effectiveness of both the spatial and temporal dependences on improving the estimation of heterogeneous snow density. It is noted that the GWNN performs inferiorly to the GRNN model with only spatial dependence, which may be caused by the sparse distribution of the stations and indirectly suggests that the temporal dependency makes a notable
contribution to improving the GRNN model. The similar accuracy differences among R, GWR, and GTWR also demonstrate the importance of the spatial and temporal dependences for snow density estimation.

Comparing the GTWNN, GWNN, and GRNN models with the GTWR, GWR, and R models, the former three models based on GRNN achieve apparently higher accuracies than the latter three models based on R. The accuracy differences mainly come from the difference between the base regression models GRNN and R, where R is a linear model and GRNN can model

nonlinear relationships. Considering that the correlation coefficients between snow density and its influencing variables are relatively weak, the results show that the nonlinear GRNN models can better overcome the weak correlations than the linear R models.

**Table 2. Accuracies of various regression models for estimating daily snow density.**

| Model | Full name of model | Slope | $R^2$ | RMSE | MAE |
|-------|--------------------|-------|-------|------|-----|
| R | Multiple linear regression | 0.783 | 0.015 | 0.060 | 0.044 |
| GWR | Geographically weighted regression | 0.069 | 0.022 | 0.143 | 0.091 |
| GTWR | Geographically and temporally weighted regression | 0.398 | 0.271 | 0.070 | 0.043 |
| GRNN | General regression neural network | 2.394 | 0.033 | 0.062 | 0.046 |
| GWNN | Geographically weighted neural network | 0.489 | 0.124 | 0.062 | 0.043 |
| GTWNN | Geographically and temporally weighted neural network | 0.906 | 0.531 | 0.043 | 0.028 |

**4.3.2 Comparison with Reanalysis Snow Density Product**

The reanalysis product ERA-5 also provides gridded daily snow density data, which is produced by comprehensively considering various influencing variables, such as snow pressure, viscosity, near surface air temperature, and wind speed (Muñoz-Sabater, 2019). We compare the snow density estimated by the GTWNN model and that in ERA-5, and the results are shown in Figure 7.

In Figure 7a, the $R^2$, RMSE, and MAE of ERA-5 snow density in the study area are 0.095, 0.061 g/cm³, and 0.047 g/cm³, respectively, the performance of which is apparently inferior to that by the GTWNN model. The ERA-5 snow density is mostly concentrated near 0.15 g/cm³, which leads to many overestimations and underestimations. Figure 7b and c further show that the snow density estimated by the GTWNN model has higher accuracies than the ERA-5 product in different snow periods and different snow cover regions.

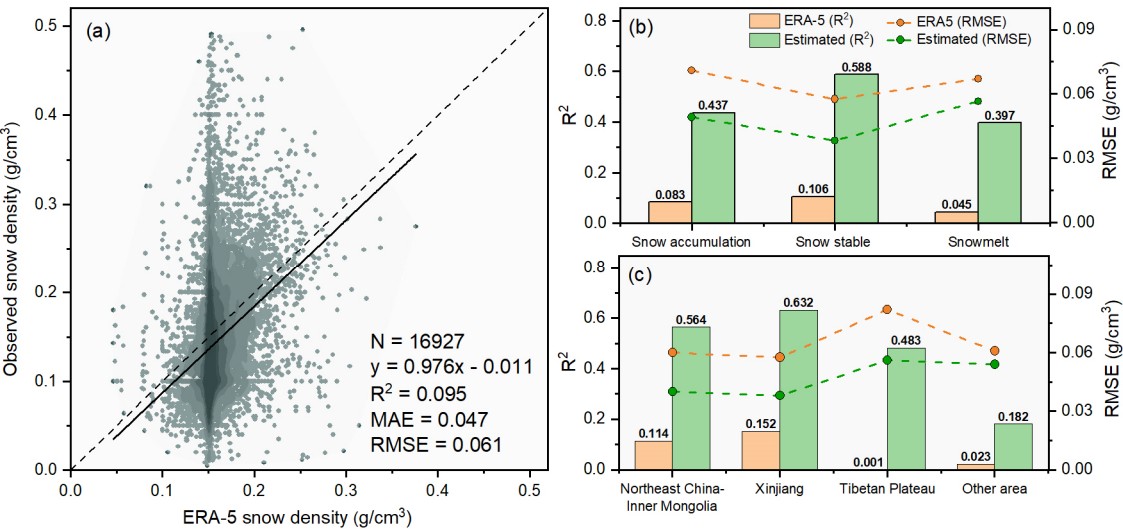

**Figure 7. Accuracies of the ERA-5 snow density product in the study area (a) and the comparison of snow density estimated by the GTWNN model with that of ERA-5 in different snow periods (b) and different snow cover regions (c).**

**4.4 Mapping of Snow Density**

**4.4.1 Spatial Distribution**

The spatial distribution of snow density in different snow periods and the entire snow season in China are mapped and shown in Figure 8a–d by calculating the average of the daily snow density estimated by the GTWNN model from 2013 to 2020. It is noted that the estimated daily snow density maps are masked by the daily snow cover product to remove the non-snow pixels

(Hao et al., 2021b). In addition, to understand the spatiotemporal heterogeneity of snow density, we also calculate the mean snow density and coefficient of variation(CV)in different snow periods and regions, as shown in Figure 8e and f.

In the snow accumulation period, the mean snow density is generally lower than 0.13 g/cm$^3$, and the difference of mean snow density in different snow cover regions is small (Figure 8e), except for the Northeast Plain and North China Plain (Figure 8a), in which the liquid water content within snow is much higher than that in other areas (Dai and Che, 2011). In the snow stable period, the mean snow density of China increases to 0.145 g/cm$^3$, especially Xinjiang and Tibetan Plateaus are above average in China (Figure 8e). The highest snow density occurs in the western Tibetan Plateau and South China (Figure 8b), which has abundant precipitation and snow density values are above 0.2 g/cm$^3$. In the snowmelt period, the mean snow density continues to increase to 0.153 g/cm$^3$, especially increasing to 0.178 g/cm$^3$ in northern Xinjiang (Figure 8e) and over 0.16 g/cm$^3$ in Changbai Mountain of Northeast China (Figure 8c). For the mean snow density of the entire snow season shown in Figure 8d and e, the mean snow density is 0.138 g/cm$^3$, 0.151 g/cm$^3$, and 0.156 g/cm$^3$ in Northeast China-Inner Mongolia, Xinjiang, and Tibetan Plateaus respectively. As shown in Figure 8d, northern Xinjiang, northwest Tibetan Plateau, and Northeast China have relatively higher snow density than Inner Mongolia and southeast Tibetan Plateau in the three major snow cover regions, which may be related to latitude, elevation, surface type (Zhong et al., 2014, 2021b).

The mean CV of snow density generally increases across China from snow accumulation period (0.170) to snowmelt period (0.192), as shown in Figure 8f. However, the CV in different snow cover regions varies apparently. It continuously decreases in Xinjiang and Tibetan Plateau from snow accumulation period to snowmelt period. However, the CV in Northeast China-Inner Mongolia achieves the lowest in the snow stable period, but that of the other area reaches the highest in the snow stable period, which may be related to the different snow classes, and the surface type, elevation, and altitude will also affect the variety of snow density. Totally in the whole snow season, Xinjiang shows the lowest CV and Northeast China-Inner Mongolia has the largest CV among the three snow cover regions.

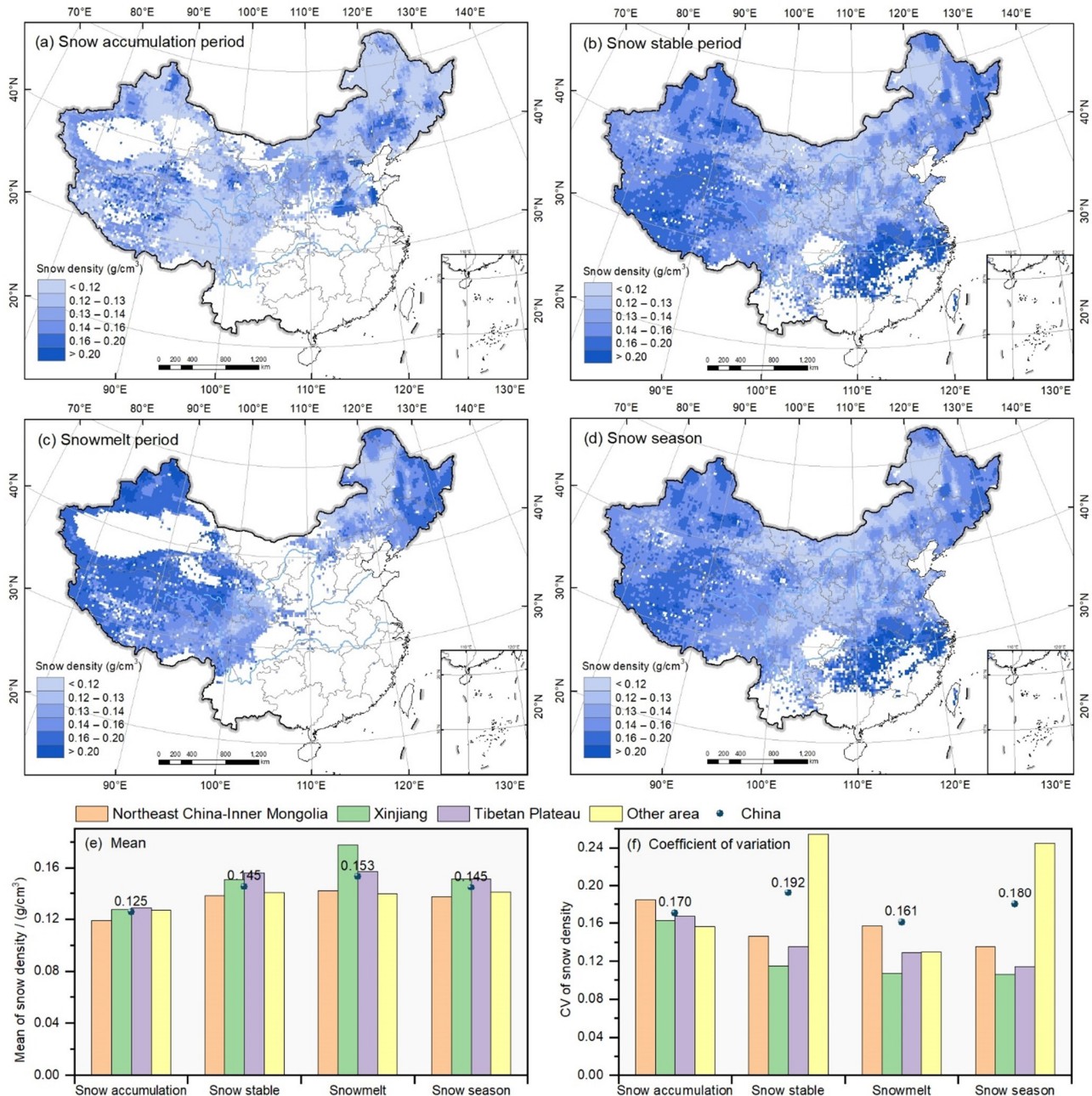

**Figure 8. Spatial distribution of mean snow density in different snow periods in China from 2013 to 2020, including the snow accumulation period (a), snow stable period (b), snowmelt period (c), and entire snow season (d), as well as the mean snow density (e) and coefficient of variation (f) in different snow periods and snow cover regions.**

### 4.4.2 Temporal Change

To reflect the monthly change in snow density in different snow cover regions, we calculate the mean snow density in each month of the snow season from January 2013 to December 2020, as shown in Figure 9a–e, as well as the monthly mean snow density of the 8 years, as shown in Figure 9f.

Figure 9a–e show that the snow density in different regions as well as the entire study area tends to increase from the start of snow accumulation to the peak and then decrease until the late snowmelt period in each year. In the snow accumulation and stable periods, snow density increases with the snow accumulation and mechanical compaction. In the early snowmelt period, snow surface melt decreases snow depth while increasing snow density via meltwater percolation, and then, most of the snow melts into water and the snow density decreases (McCreight and Small, 2014).

However, the snow density fluctuations appear different over time and space. Specifically, the months with the maximum and minimum snow density are various in different regions, which may be related to the climatic conditions. The monthly changes in Xinjiang and Tibetan Plateau are similar and apparently different from those in Northeast China-Inner Mongolia, which is because the temperature and gradient between snow and atmosphere is small in Northeast China (Ebner et al., 2016), with low air temperature and vapor pressure in the snow stable period (Ji et al., 2017). In addition, snow cover is relatively shallow, and the metamorphism caused by the compaction is not significant (Yang et al., 2020), which allow the snow density in Northeast China-Inner Mongolia to fluctuate less during the seasonal changes. However, the seasonal evolution of snow density is obvious at the high altitudes and elevation areas of Xinjiang and Tibetan Plateau, possibly because of the relatively high water vapor (Ji et al., 2017) and the temperature cycling between day and night that accelerates snow metamorphism (Ebner et al., 2016).

In addition, the monthly mean snow density from the estimated daily snow density map in Figure 9f shows a similar pattern with that from the ground observations in Figure 3b, which further demonstrates the effectiveness of the proposed GTWNN model for snow density estimation.

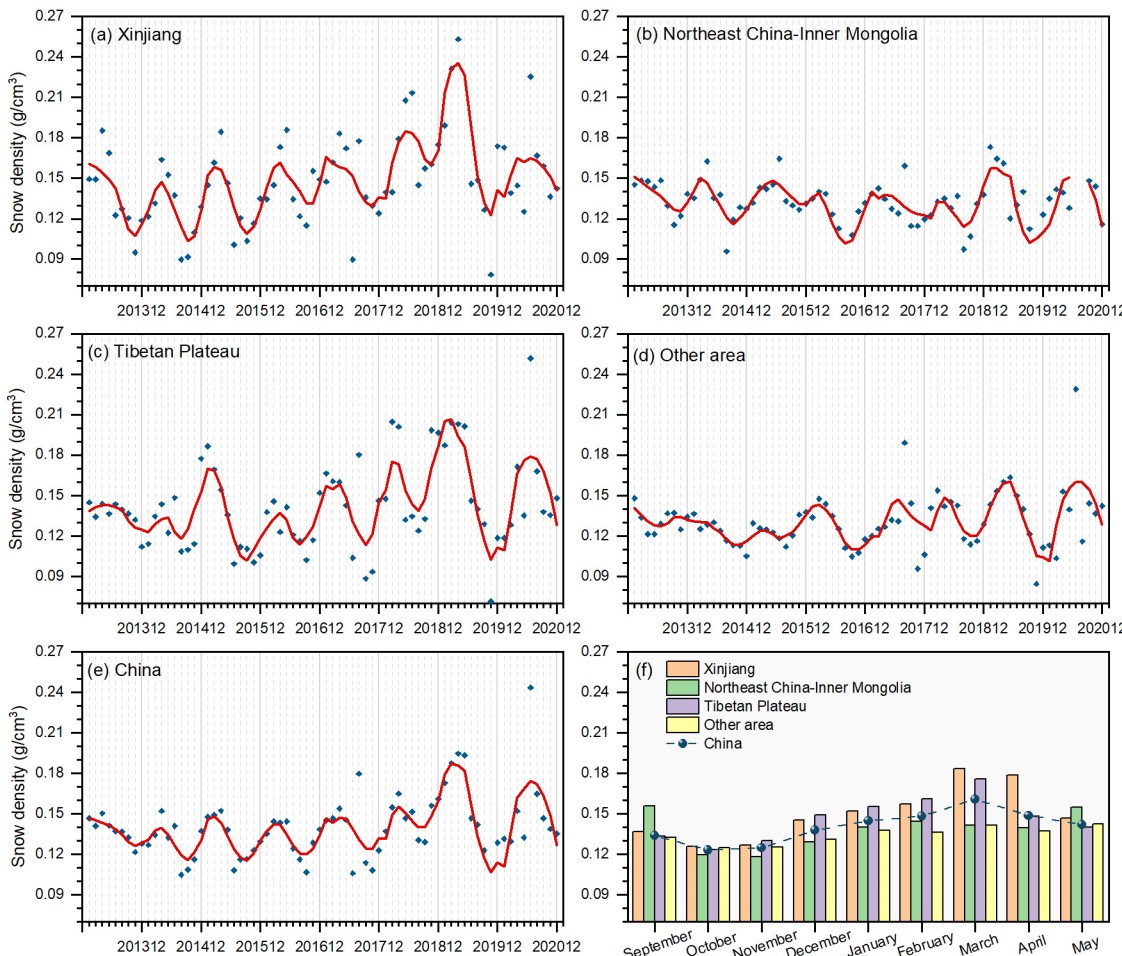

Figure 9. Mean snow density in each month of the snow season from January 2013 to December 2020 in different snow cover regions, including Xinjiang (a), Northeast China-Inner Mongolia (b), Tibetan Plateau (c), other area (d), and the entire study area (e), as well as the monthly mean snow density of the eight years (f).

## 5 Discussion

### 5.1 Essential Issues of Constructing and Applying GTWNN Model

The constructed GTWNN model achieves daily snow density mapping by integrating a variety of influencing variables with the support of remote sensing, ground observation, and reanalysis data. Even though the validated accuracies are acceptable,

it is also necessary to address three essential issues for constructing and applying the model: (1) the weak correlation between influencing variables and snow density, (2) the model evaluation and parameter estimation, and (3) the relation to the state of snow.

420         For the first issue, we found that the influencing variables have relatively limited explanatory abilities for estimating snow density, as indicated by the weak correlations in Figure 4, which may be an important reason for the low accuracy of the GTWNN model for snow density estimation with an $R^2$ of 0.531. For example, Li et al. (2020) established a GTWNN model for estimating ground-level $PM_{2.5}$ with the input of satellite-derived aerosol optical depth (AOD) and meteorological data, and the model achieved an $R^2$ value of 0.80, which may be related to the higher linear correlation between AOD and $PM_{2.5}$ with

an $R^2$ of 0.75 (Xin et al., 2016). Considering the weak correlations between snow density and its influencing variables, it could be challenging to achieve very high estimation accuracy for snow density.

        In addition, the accuracy of influencing variables would also affect the GTWNN model estimation accuracy. We downloaded the instantaneous near surface (2 m) air temperature and precipitation from the China meteorological forcing dataset (CMFD), with a spatial resolution of 0.1° for comparison. CMFD is the high spatial-temporal resolution gridded near-

surface meteorological dataset in China, which was made through fusing remote sensing products, reanalysis datasets, and in-situ station data (He et al., 2020). Since CMFD only provides data until 2018, we use CMFD data to replace the temperature and precipitation data of ERA-5, and the accuracies of the models with different influencing variables from 2013 to 2017 are shown in Table 3. The accuracies of the new model with CMFD are slightly higher than those of original model indicated by $R^2$, but the RMSE and MAE remain the same. However, considering the high spatiotemporal resolution and rich variables,

especially the temporal coverage of ERA-5 data (1950–), we finally choose the ERA-5 data in our study.

**Table 3. Accuracy comparison of estimated snow density with different sources of influencing variables.**

| Year | GTWNN model with ERA-5 data | | | GTWNN model with CMFD data | | |
|---|---|---|---|---|---|---|
| | $R^2$ | RMSE | MAE | $R^2$ | RMSE | MAE |
| 2013 | 0.499 | 0.041 | 0.025 | 0.495 | 0.041 | 0.025 |
| 2014 | 0.521 | 0.040 | 0.024 | 0.531 | 0.039 | 0.024 |
| 2015 | 0.473 | 0.042 | 0.027 | 0.482 | 0.042 | 0.027 |
| 2016 | 0.560 | 0.038 | 0.023 | 0.575 | 0.037 | 0.023 |
| 2017 | 0.591 | 0.040 | 0.023 | 0.587 | 0.040 | 0.023 |
| Overall | 0.529 | 0.040 | 0.024 | 0.534 | 0.040 | 0.024 |

        For the second issue, there are three essential parameters in the GTWNN model, including the scale factor $\varphi$ and the spatiotemporal bandwidth $h_{ST}$, and the $spread$. Especially for the bandwidth. When we choose the adaptive bandwidth regime $h_{ST} = 6$, the $R^2$ of the estimated snow density could achieve a higher value of 0.542. However, there will be abnormal block

patterns in the snow density map caused by the limited number of samples. Specifically, in a small area constrained by $h_{ST} = 6$, the same samples can be selected for estimating the snow density of different nearby grids. Since the spatiotemporal weights and the weights of influencing variables are too similar, it results in the block phenomenon over areas composed of grids with very similar snow density. Therefore, even though a relatively smaller $h_{ST}$ could achieve a higher $R^2$, we have to increase the bandwidth $h_{ST} = 8$ to avoid the abnormal block patterns that are inconsistent with our common sense.

445         For the third issue, it is noted that the accuracy of the snow density estimation model is closely related to the stability and amount of snow. The $R^2$ values in the three major snow cover regions are apparently higher than those in the other areas in China because of the more stable and larger amount of snow, as shown in Figure 5. In addition, the accuracy differences in different months show that the GTWNN model achieves higher performance when snow is more stable, such as the snow stable period, the late snow accumulation period, and the early snowmelt period. Hence, when applying the GTWNN model,

if the snow is stable and plentiful, the estimated snow density would be more credible. In contrast, if the snow changes rapidly, distributes sparsely, or the observation difficulty increases, such as in the early snow accumulation period and the late snowmelt period, the estimated snow density would be less credible and need to be used with caution.

## 5.2 Advantages and Limitations

Even though the ground measurement accuracy continues to improve by advanced measurement instruments (Hao et al.,
2021a), it is limited to obtaining the large-scale spatial distribution of snow density, and gridded snow density data are urgently needed. To obtain the gridded snow density, snow parameters such as snow depth (Jonas et al., 2009; McCreight and Small, 2014) and meteorological data such as temperature and wind speed (Helfricht et al., 2018; Judson and Doesken, 2000; Valt et al., 2018) were used to estimate snow density mainly by using a linear regression model. The linear regression model is globally oriented and thus cannot effectively deal with the spatiotemporal heterogeneity of snow density. Accordingly, previous studies
mostly achieve snow density estimation in regions that are not very large in size. The constructed GTWNN in this study considers the spatial and temporal dependences of snow density, which allows it to effectively deal with the spatiotemporal heterogeneity of snow density and thus hold the potential of being applied to large-scale areas, as demonstrated by the apparently higher accuracies than the linear regression model in our study area. In addition, it is important to overcome the weak correlation between snow density and its influencing variables to improve the estimation accuracy. Accordingly, we
make two efforts in the GTWNN model. First, 20 influencing variables are integrated for the estimation with the support of multisource data. Second, the adopted GRNN model could overcome the nonlinear relationship between snow density and its influencing variables.

It is noted that the GTWNN model is a spatiotemporal interpolation model based on the observed snow density, and the confidence of the snow density map produced by the GTWNN model is still constrained by the distribution of the observation
stations, even though the model is able to achieve relatively high accuracy in regions with sparse stations, e.g., the $R^2$ of 0.517 on Tibetan Plateau. Since there are few observations, especially in northwest Tibetan Plateau, the confidence of the estimated snow density in this region is still not clear. On the one hand, we expect new observations in the near future for both estimation and validation. On the other hand, it is of great potential to further develop a snow density prediction model without the dependence of observed snow density for model inference.

## 6 Conclusion

A GTWNN model was constructed for snow density estimation and achieved daily snow density mapping from 2013 to 2020 in China with the support of remote sensing, ground observation, and reanalysis data. The GTWNN model has two advantages: (1) considering the spatiotemporal heterogeneity of snow density, and (2) addressing the weak and nonlinear relationship as well as the involvement of a variety of snow, meteorological, topographic, and vegetation variables. The individual correlations
between snow density and 20 influencing variables are very week, with the maximum average correlation coefficient of only 0.123, and it is found that the vegetation variable LAI_HV, meteorological variable TP, snow variable SD, and topographic variables have a relatively close relationship to snow density. The GTWNN model achieves an $R^2$ of 0.531, RMSE of 0.043 g/cm$^3$, and MAE of 0.028 g/cm$^3$ in China validated by 10-fold cross validation, which are apparently better than those of the other five regression models and the ERA-5 snow density product. The comparison results further demonstrate the importance
of addressing the spatiotemporal heterogeneity for snow density estimation. The performance of the GTWNN model is also demonstrated to be closely related to the state and amount of snow, in which more stable and plentiful snow would result in higher snow density estimation accuracy. With the benefit of the produced daily snow density map, we obtain knowledge of the spatiotemporal pattern of snow density in different snow periods and snow cover regions in China, and the CV results show that spatial heterogeneity of snow density in Northeast China-Inner Mongolia is the most obvious in three major snow cover

regions, and the least obvious in Xinjiang. The proposed GTWNN model has the potential to be used for large-scale snow density mapping because of the two advantages described above but with limitations to the distribution of the observation stations. Future work should focus on extending the model to other areas and longer time series as well as developing snow density prediction models.

**Data availability**

Our research mainly uses three types of data, including the ground observation data, satellite remote sensing data, and reanalysis data. The daily snow depth and snow pressure measurements are collected from the China Meteorological Administration (CMA) and the National Cryosphere Desert Data Center (http://www.ncdc.ac.cn), we are not authorized to redistribute part of these data without permission. The remote sensing data we used are all open source data. The products of snow variables are available at the National Cryosphere Desert Data Center (http://www.ncdc.ac.cn), including the daily snow

albedo, snow depth, and snow cover area from 2013 to 2020, and the Shuttle Radar Topography Mission (SRTM) digital elevation model (DEM) can be downloaded at http://www.dsac.cn/DataP. The reanalysis data are collected from the ECMWF ERA-5 land hourly dataset (https://cds.climate.copernicus.eu), and the MODIS land vegetation cover classification product (MCD12Q1) can be download at https://modis.gsfc.nasa.gov/.

**Author contribution**

Xueliang Zhang and Pengfeng Xiao formulated the study goals; Huadong Wang, Zhaojun Zheng, Liyun Dai, and Tao Che performed the data curation; Huadong Wang and Xueliang Zhang analyzed the data and wrote the manuscript draft; Pengfeng Xiao, Tao Che, Zhaojun Zheng, Liyun Dai, and Wenbo Luan reviewed and edited the manuscript.

**Competing interests**

The authors declare that they have no conflict of interest.

**Acknowledgments**

This study is supported by the Special Subject of National Science and Technology Basic Resources Investigation (Grant No. 2017FY100503), the National Natural Science Foundation of China (Grant No. 42171307), the Fundamental Research Funds for the Central Universities (Grant No. 020914380095), and the High-level Innovation and Entrepreneurship Talents Introduction Program of Jiangsu Province of China. The authors would like to thank the editors and reviewers for their

constructive comments to improve the paper.

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
