# Peer review of "Towards Large-Scale Daily Snow Density Mapping with Spatiotemporally Aware Model and Multi-Source Data"

_The Cryosphere, 2022_

## Referee Comment (RC2)

Snow density plays a critical role in the estimation of snow water equivalent (SWE). Predicting a temporally and spatially variant snow density is not trivial and is usually assume constant for SWE estimates. This study presents a geographically and temporally weighted neural network (GTWNN) model to predict daily snow density across China. This work relies on empirical relations with influencing variables and machine learning algorithm, to predict density over time and space. This work proposes a great way to map snow density over China, but further clarifications are needed before publications.

In general, no physical understanding of snow density with influencing variable was explored or used in the modelling. This method relies purely on empirical relations. Not those empirical relations cannot be used but perhaps adding a bit more physical understanding in the variables selection or using a physical model at the regional scale could improve this work.

Specific comments:

L52-56 This paragraph needs more on how topography and vegetation influence snow density. It might also be also useful to define the scale a which these processes operate relative to this work.

L71 This is true but maybe used them at the regional scale to add a physical basis of energy exchange in the snowpack.

L90 It is stated "to understand how the influencing variables affect snow density estimation". How was this address in the study?

L120 More is needed here on how the topographic parameters were estimated. Was the mean of all pixels at 30m resolution used to estimate the elevation? Could the standard deviation or other statistical parameters of sub pixel variability be used?

Section 3.2 It is not clear how the model is evaluated… against ground observations? It says in the objectives that daily snow density mapping is achieve by integrating satellite, ground and reanalysis data. One or two sentences are needed here to clarify which is used for what and how the model is trained and validated.

Figure 4 Again, how was it trained and validated. Can you define the dataset percentage used for training and validation? Was it trained on some years and evaluated on the remaining years and same for the region?

Section 4.2.3 Other methods than Pearson correlation factor can be used to investigate the importance of influencing variables. This only indicates a correlation. I would suggest using a permutation importance-based method or an impurity importance from a tree classifier. Maybe it would give better insight on the variables.

Section 4.3.2 What does this section adds to the manuscript. Does it relate to the objectives? Also, most of the influencing variables come from the ERA-5 reanalysis dataset. Does it affect the results?

Line 363 It is stated that weak correlations exist between snow density and the influencing variables chosen for the predictive model. Could a physical snowpack model (ISNOBAL, CROCUS or SNOWPACK) be used for the 4 different regions (not all pixel) to try to add a physical base to the prediction that is mostly empirical through weak correlations at the moment?

Line 389 The GTWNN can deal with spatiotemporal heterogeneity but how about temporal and spatial transferability of the model in the training/validation?

Line 402 How would that be achieved? Using a physical model?

---

## Author Comment (AC1)

**Response to reviewer**

**May 3rd, 2022**

Tc-2022-45 "Towards Large-Scale Daily Snow Density Mapping with Spatiotemporally Aware Model and Multi-Source Data"

We would like to thank the reviewer for your constructive comments, which helped to substantially improve the manuscript. Below we will address each concern in a point-by-point answer:

- **Bold: comments of the reviewer**
- Regular: answer of the authors
- *Italics or* red words *: changes to the initial manuscript*

**Comments by reviewer:**

**Snow density plays a critical role in estimating water resources and predicting natural disasters such as floods, avalanches, and snowstorms. A GTWNN model was constructed for snow density estimation and achieved daily snow density mapping from 2013 to 2020 in China with the support of remote sensing, ground observation, and reanalysis data. This study provides important spatiotemporal parameters for snow cover hydrology and other aspects. The main suggestions and opinions are as follows:**

**1. L115, "Based on the SCA data, the snow cover duration (SCD) is calculated to account for the impact of gravity on snow density", How to understand that snow density is affected by gravity, and what does it have to do with SCD?**

*Response from Authors:*

The authors greatly thank for the question. The aim of calculating the snow cover duration (SCD) is to account for the impact of snow duration on snow density rather than the gravity. Several models considered the effect of inputting time series on snow density, for example for seasonal (Sturm et al., 2010) and biweekly (Jonas et al., 2009) timescales, which indicates that the accumulated snow cover days cannot be ignored. We made changes in Subsection 2.2 Satellite and Reanalysis Data.

*Revised subsection 2.2 Satellite and Reanalysis Data*

"Based on the SCA data, the snow cover duration (SCD) is calculated to account for the impact of snow duration on snow density."

**2. "Spatiotemporally Aware Model" in the title is not mentioned in the manuscript and should be explained.**

*Response from Authors:*

Thanks for the comment on details. The spatiotemporally aware model represents the geographically and temporally weighted neural network (GTWNN) model, which consists of geographically and temporally weighted (GTW) model to capture spatiotemporal heterogeneity and a generalized regression neural network (GRNN) to deal with the weak and nonlinear relationships between snow density and its influencing variables. We explained it in the revised manuscript.

*Revised subsection 3.1 GTWNN Model*

The schematic of the GTWNN model is shown in Figure 2, which is a spatiotemporally aware model composed of a geographically and temporally weighted (GTW) model to capture spatiotemporal heterogeneity and a generalized regression neural network (GRNN) to deal with the weak and nonlinear relationships between snow density and its influencing variables

**3. Whether the lack of observation data in 2019-2020 is related to the epidemic, making it impossible to conduct**

**a large number of observations.**

40    *Response from Authors:*

Thank you very much for this comment. The smaller numbers of observation data in 2019–2020 is because of the lack of snow pressure data recorded by the meteorological stations. According to the regulations, a lot of stations only observed snow depth and did not observe snow pressure, which makes it impossible to calculate snow density and the number of observations is significantly reduced in 2019–2020.

45    **4. The verification result in Fig.4 is that all the data as a whole is added to the training model, or is the training divided by region and month? Is it the 10-fold validation result of the trained model? Please explain further**

*Response from Authors:*

Thank you very much for the question. All data are divided into 8 parts by year from 2013 to 2020, and each part is used for training by the 10-fold validation, which means all the samples (each year) are divided into 10 equal folds
50    randomly, nine folds are exploited for the model fitting, and the one remaining fold is used for the validation, the above step is repeated 10 times. Finally, we built the models of each year and obtain the estimated snow density of all data, which are compared with the observed snow density to verity the model accuracy. We added the explanation in the revised manuscript.

The reason for training by year is on the consideration of the spatiotemporal heterogeneity of snow density and the
55    different importance of influencing variables in different years. Accordingly, the stepwise regression analysis method is used to select significant variables for the GTWNN model in each year. Motivated by the raised question, we also tried to train the GTWNN models by region and by month with the 10-fold validation, as shown in Table S1. Comparing the models trained by month, region, and year, the former two models achieve relatively higher $R^2$ than the model by year, but the RMSE and MAE remain unchanged. Hope our efforts have addressed the major concerns.

60    Table S1. Accuracies of various methods for estimating daily snow density.

| Method | Slope | $R^2$ | RMSE | MAE |
|--------|-------|-------|------|-----|
| Month | 0.988 | 0.521 | 0.043 | 0.028 |
| Region | 0.983 | 0.518 | 0.043 | 0.028 |
| Year | 0.986 | 0.515 | 0.043 | 0.028 |

*Revised subsection 3.2 Model Evaluation*
The GTWNN model is constructed separately for each year with the consideration of the snow variety in different years. The stepwise regression analysis method is used to filter the influencing variables by performing the F test ($\alpha < 0.05$) and T test ($\alpha < 0.1$) to select significant variables for the GTWNN model in each year. There are three essential parameters in each
65    GTWNN model, including the spatiotemporal bandwidth $h_{ST}$ and the scale factor $\varphi$ of the GTW model and the *spread* of GRNN model. The 10-fold cross-validation technique is adopted for determining the optimal parameters (Fotheringham et al., 2003; Rodriguez et al., 2010); that is, all the collected samples are randomly divided into 10 folds, nine folds are exploited for the model fitting, and one fold is used for the validation. Finally, a scale factor $\varphi$ of 0.01, *spread* of 0.5, and an adaptive bandwidth regime $h_{ST}$ of 8 with the same best model performance are obtained in different year.

70    **5. L200, Does the reason for the lower accuracies in Northeast China-Inner Mongolia consider the effect of different underlying surfaces on snow density? Forests and farmland in the Northeast, and grasslands in Inner Mongolia may have very different effects on snow density.**

*Response from Authors:*

The authors greatly agree with this suggestion that the different underlying surface will affect the estimation of

75 snow density. Actually, we have considered the influence of underlying surfaces by using the leaf area index of high vegetation (LAI_HV), which is the one-half of the total green leaf area per unit horizontal ground surface area for high vegetation type. The variable LAI_HV has the largest average correlation coefficient among all the variables and is selected for modeling in most of the years, except for 2019, as shown in Figure 6.

    Moreover, according to the advice, we downloaded the MODIS land vegetation classification product

80 (MCD12Q1_V6) with a spatial resolution of 500 m and resample them to 25 km to further explore the influence of different land cover on snow density.

    Based on the stepwise regression analysis method, the MCD12Q1 is selected as the significant variables only in 2013 and 2017, and the Pearson correlation coefficients between snow density and MCD12Q1 are 0.0446 and -0.0556, respectively. The correlation relationship is relatively weak compared with other influencing variables, as shown in

85 Figure 6. We calculated the accuracy of GTWNN model after inputting the MCD12Q1 into the models as an influencing variable, which is slightly improved in comparison with the original GTWNN model without inputting MCD12Q1, with $R^2$ 0.521, RMSE 0.043 g/cm$^3$, and MAE 0.028 g/cm$^3$, as shown in Table S2. In addition, it is noted that the accuracy improvement by inputting MCD12Q1 in Northeast China-Inner Mongolia and Tibetan Plateau is higher than that in Xinjiang.

90    Therefore, both the high correlation value of the LAI_HV variable and the improvement by inputting MCD12Q1 demonstrate the importance of underlying surfaces for estimating snow density, especially for the Northeast China-Inner Mongolia. Accordingly, the lower accuracies in Northeast China-Inner Mongolia would mainly be caused by the more forest cover and less stable snow cover in Northeast China-Inner Mongolia than in Xinjiang (See revised subsection 4.2.1 Accuracy of GTWNN Model in Different Regions). Hope our efforts have addressed the major concerns.

[Figure]

95

**Figure 6. Correlation coefficient between snow density and its influencing variables selected by the stepwise regression method in each year (a), and the average of the absolute value of the correlation coefficient and the number of selections within these years (b).**

**Table S2. Impact of MCD12Q1 on accuracies of estimated snow density.**

| Model | Region | Slope | $R^2$ | RMSE | MAE |
|---|---|---|---|---|---|
| Original GTWNN model | China | 0.986 | 0.515 | 0.043 | 0.028 |
| | Xinjiang | 1.016 | 0.632 | 0.038 | 0.025 |
| | Northeast China-Inner Mongolia | 1.018 | 0.564 | 0.040 | 0.025 |
| | Tibetan Plateau | 0.926 | 0.483 | 0.056 | 0.040 |
| | Other areas | 0.793 | 0.182 | 0.054 | 0.038 |
| GTWNN model with the input of MCD12Q1 | China | 0.968 | 0.521 | 0.043 | 0.028 |
| | Xinjiang | 1.011 | 0.634 | 0.038 | 0.025 |
| | Northeast China-Inner Mongolia | 0.997 | 0.575 | 0.039 | 0.024 |
| | Tibetan Plateau | 0.934 | 0.497 | 0.055 | 0.040 |
| | Other areas | 0.740 | 0.180 | 0.054 | 0.038 |

**6. The reasons for the slightly lower accuracy in the snow melting and accumulation periods are not only the rapid changes in the snow density itself, and insufficient sampling in observation time and space, but also because the snow accumulation in the early stage of snow accumulation is less, and the water content when the snow melts Therefore, the observation is more difficult, and the observation error is relatively large.**

*Response from Authors:*

Thanks for providing the professional suggestion. These comments consider the snow characteristics during the snow accumulation and melt periods, which lead to increased observation difficulty and measurement error, and thus can more comprehensively explain the reason for the slightly lower accuracy in these periods. We added the reason in our revised manuscript.

*Revised subsection 4.2.2 Accuracy of GTWNN Model in Different Months*

It is noted that the observation error is also cannot be ignored, which may be caused by the less snow in the early stage of snow accumulation period, or the large water content when the snowmelt period, making the observation more difficult.

*Revised subsection 5.1 Essential Issues of Constructing and Applying GTWNN Model*

In contrast, if the snow changes rapidly, distributes sparsely, or the observation difficulty increases, such as in the early snow accumulation period and the late snowmelt period, the estimated snow density would be less credible and need to be used with caution.

**7. The verification result of the snow density of ERA5 is worse than that of the model in this paper, but many parameters of ERA5 are used in the machine learning model of this paper, so the accuracy of these parameters, if there is also a large error, will not affect the final model accuracy?**

*Response from Authors:*

Thanks for this comment. The reason for choosing the ERA-5 data is that the high spatiotemporal resolution and rich variables compared to other reanalysis data, with a spatial resolution of 0.1° and a temporal resolution of one hour. The near-surface meteorological state and flux fields, including the air temperature, wind speed, surface pressure, and total precipitation, are corrected for the altitude differences and have improved quality (Muñoz-Sabater et al., 2021). The reanalysis can provide an estimation of the meteorological gridded dataset by assimilating various observations into the forecast model system (Dee et al., 2014). The ECMWF ERA-5 land hourly dataset, as any other simulation, provides estimates which have some degree of uncertainty.

To verify whether the accuracy of the influencing variables affect the final model accuracy, we downloaded the instantaneous near surface (2 m) air temperature and precipitation from the China meteorological forcing dataset (CMFD), with a spatial resolution of 0.1° for comparison. CMFD is the high spatial-temporal resolution gridded near-

130  surface meteorological dataset in China, which was made through fusion of remote sensing products, reanalysis datasets and in-situ station data (He et al., 2020). Since CMFD only provides data until 2018, we use CMFD data to replace the temperature and precipitation data of ERA-5, and the accuracies of the models with different influencing variables from 2013 to 2017 are shown in Table S3.

The accuracies of new model with CMFD are slightly higher than those of original model indicated by $R^2$, but the
135  RMSE and MAE remain the same, which indicates that the accuracy of the influencing variables will affect the model accuracy, but the temperature and precipitation data of ERA-5 are comparable to that of CMFD for driving our model. It is also noted that the differences of influencing variables will affect the stepwise regression analysis results, making the significant variables and the correlation coefficient between snow density and the selected influencing variables changed each year, which may also affect the final model accuracy.

140  According to the above results, we can conclude that the accuracy of influencing variables would affect the final model accuracy. Even though the accuracy of ERA-5 snow density worse than ours, the temperature and precipitation data of ERA-5 achieve comparable performance with CMFD data for driving our model. In addition, considering the high spatiotemporal resolution and rich variables, especially the temporal coverage of ERA-5 data (1950–), we finally choose the ERA-5 data in this study.

145  We added discussion about the impact of low accuracy of influencing variables on snow density estimation in the revised manuscript. Hope our efforts have addressed the major concerns.

*Revised subsection 5.1 Essential Issues of Constructing and Applying GTWNN Model*

In addition, the accuracy of influencing variables would also affect the GTWNN model estimation accuracy.

**Table S3. Accuracy of estimated snow density with different influencing variables.**

| Year | Original model | | | New model (CMFD) | | |
|---|---|---|---|---|---|---|
| | $R^2$ | RMSE | MAE | $R^2$ | RMSE | MAE |
| 2013 | 0.477 | 0.042 | 0.027 | 0.497 | 0.041 | 0.027 |
| 2014 | 0.483 | 0.041 | 0.026 | 0.483 | 0.041 | 0.026 |
| 2015 | 0.448 | 0.043 | 0.029 | 0.442 | 0.043 | 0.028 |
| 2016 | 0.531 | 0.039 | 0.025 | 0.535 | 0.039 | 0.025 |
| 2017 | 0.533 | 0.042 | 0.026 | 0.546 | 0.042 | 0.026 |
| Overall | 0.497 | 0.041 | 0.027 | 0.503 | 0.041 | 0.027 |

150

**References**

Dee, D. P., Balmaseda, M., Balsamo, G., Engelen, R., Simmons, A. J., and Thépaut, J.-N.: Toward a Consistent Reanalysis of the Climate System. Bulletin of the American Meteorological Society, 95(8), 1235–1248, https://doi.org/10.1175/BAMS-D-13-00043.1, 2014.

He, J., Yang, K., Tang, W., Lu, H., Qin, J., Chen, Y. and Li, X.: The first high-resolution meteorological forcing dataset for land process
155  studies over China, Scientific Data, 7, 25, https://doi.org/10.6084/m9.figshare.11558439, 2020.

Jonas, T., Marty, C., and Magnusson, J.: Estimating the snow water equivalent from snow depth measurements in the Swiss Alps, Journal of Hydrology, 378, 161–167, https://doi.org/10.1016/j.jhydrol.2009.09.021, 2009.

Muñoz-Sabater, J., Dutra, E., Agustí-Panareda, A., Albergel, C., Arduini, G., Balsamo, G., Boussetta, S., Choulga, M., Harrigan, S., Hersbach, H., Martens, B., Miralles, D.G., Piles, M., Rodríguez-Fernández, N.J., Zsoter, E., Buontempo, C., Thépaut, J.-N.: ERA5-Land: a state-
160  of-the-art global reanalysis dataset for land applications, Earth System Science Data, 13, 4349–4383. https://doi.org/10.5194/essd-13-4349-2021, 2021.

Sturm, M., Taras, B., Liston, G. E., Derksen, C., Jonas, T., and Lea, J.: Estimating snow water equivalent using snow depth data and climate classes, Journal of Hydrometeorology., 11, 1380–1394, https://doi.org/10.1175/2010JHM1202.1, 2010

165  **Our authors greatly appreciate the advices and suggestions from the reviewer, and we tried to further improve our work accordingly, hope the revisions could successfully address the raised concerns. Many thanks again!**

---

## Author Response (AR1)

**Response to reviewers**

Tc-2022-45 "Towards Large-Scale Daily Snow Density Mapping with Spatiotemporally Aware Model and Multi-Source Data"

The authors greatly thank all editors and reviewers for their efforts of reviewing our paper and the constructive comments. The comments are valuable in helping us improve our work. Below we will address each concern in a point-by-point answer:

- **Bold: comments of the reviewer**
- Regular: answer of the authors
- *Italics or* red words*: changes to the initial manuscript*

We hope that the revised manuscript has successfully addressed the raised concerns. Thank you once again for your attention to our paper.

**Comments & Suggestions by Community #1**

**Snow density plays a critical role in estimating water resources and predicting natural disasters such as floods, avalanches, and snowstorms. A GTWNN model was constructed for snow density estimation and achieved daily snow density mapping from 2013 to 2020 in China with the support of remote sensing, ground observation, and reanalysis data. This study provides important spatiotemporal parameters for snow cover hydrology and other aspects. The main suggestions and opinions are as follows:**

**1. L115, "Based on the SCA data, the snow cover duration (SCD) is calculated to account for the impact of gravity on snow density", How to understand that snow density is affected by gravity, and what does it have to do with SCD?**

*Response from Authors:*

The authors greatly thank for the question. The aim of calculating the snow cover duration (SCD) is to account for the impact of snow duration on snow density rather than the gravity. Several models considered the effect of inputting time series on snow density, for example for seasonal (Sturm et al., 2010) and biweekly (Jonas et al., 2009) timescales, which indicates that the accumulated snow cover days cannot be ignored. We made changes in Section 2.2 Satellite and Reanalysis Data.

**2.2 Satellite and Reanalysis Data**

Based on the SCA data, the snow cover duration (SCD) is calculated to account for the impact of snow duration on snow density.

**2. "Spatiotemporally Aware Model" in the title is not mentioned in the manuscript and should be explained.**

*Response from Authors:*

Thanks for the comment on details. The spatiotemporally aware model represents the geographically and temporally weighted neural network (GTWNN) model, which consists of geographically and temporally weighted (GTW) model to capture spatiotemporal heterogeneity and a generalized regression neural network (GRNN) to deal with the weak and nonlinear relationships between snow density and its influencing variables. We explained it in the revised manuscript.

**3.1 GTWNN Model**

The GTWNN model is a spatiotemporally aware model composed of a geographically and temporally weighted (GTW) model to capture spatiotemporal heterogeneity and a generalized regression neural network (GRNN) to deal with the weak and nonlinear relationships between snow density and its influencing variables, including the meteorological variables, topographical variables, vegetation variables, and snow variables, which could be expressed as shown in Eq. (1), and its schematic is shown in Figure 2.

**3. Whether the lack of observation data in 2019-2020 is related to the epidemic, making it impossible to conduct a large number of observations.**

*Response from Authors:*

Thank you very much for this comment. The smaller numbers of observation data in 2019–2020 is because of the lack of snow pressure data recorded by the meteorological stations. According to the regulations, a lot of stations only observed snow depth and did not observe snow pressure, which makes it impossible to calculate snow density and the number of observations is significantly reduced in 2019–2020.

**4. The verification result in Fig.4 is that all the data as a whole is added to the training model, or is the**

**training divided by region and month? Is it the 10-fold validation result of the trained model? Please explain further**

*Response from Authors:*

Thank you very much for the question. Originally, all data are divided into 8 parts by year from 2013 to 2020, and each part is used for training by the 10-fold validation, which means all the samples (each year) are divided into 10 equal folds randomly, nine folds are exploited for the model fitting, and the one remaining fold is used for the validation. The above steps are repeated 10 times. Finally, we built the models of each year and obtain the estimated snow density of all data, which are compared with the observed snow density to verity the model accuracy. We added the explanation of 10-fold validation in the revised Section 3.2 Parameter Determination and Model Evaluation Method.

The reason for training by year is on the consideration of the different importance of influencing variables in different years. Accordingly, the stepwise regression analysis method is used to select significant variables for the GTWNN model in each year. Motivated by the raised question, we also tried to train the GTWNN models by region and by month with the 10-fold validation, and the model accuracies are shown in Table S1. Comparing the models trained by month, region, and year, the former two models achieve relatively higher $R^2$ than the model by year, but the RMSE and MAE remain unchanged. However, we found that the variables selected by the stepwise regression method are fewer in some months or regions, which may affect the results of snow density map. Therefore, we tried to input all influencing variables into GTWNN model, the $R^2$ is also higher than the model by year, as shown in Table S1.

Finally, we choose to input all influencing variables into the GTWNN model in our revised manuscript, and to calculate the Pearson correlation coefficient between snow density and all the influencing variables in different months rather than in each year, to better understand the relationship between snow density and different influencing variables, as shown in the revised Section 4.2.1 Relationship between Snow Density and Its Influencing Variables. Hope our efforts have addressed the major concerns.

**Table S1. Accuracies of various methods for estimating daily snow density.**

| Method | Slope | $R^2$ | RMSE | MAE |
|---|---|---|---|---|
| Month | 0.961 | 0.531 | 0.043 | 0.028 |
| Region | 0.908 | 0.531 | 0.043 | 0.028 |
| Year | 0.946 | 0.520 | 0.043 | 0.028 |
| All variables | 0.906 | 0.531 | 0.043 | 0.028 |

*3.2 Parameter Determination and Model Evaluation Method*

To evaluate the model performance as well as to determine the optimal parameters, the 10-fold cross-validation technique is adopted (Fotheringham et al., 2003; Rodriguez et al., 2010); that is, all the collected samples are randomly divided into 10 folds, nine folds are exploited for the model fitting, and one fold is used for the validation. The above steps are repeated 10 times so as to evaluate the model performance on each fold of the validation samples.

*4.2.1 Relationship between Snow Density and Its Influencing Variables*

The Pearson correlation coefficient between snow density and its influencing variables is calculated to indicate the importance of the variables in each month, as shown in Figure 4a, where September and May are not involved because of the small number of ground observations. The influencing variables and the corresponding correlation coefficient values are various in different months because of the heterogeneity of snow. In addition, we calculate the average value from October to April for the positive and negative correlation coefficients, respectively, to indicate the importance of each influencing variable for snow density. We also count the number of months with positive or negative correlations and mark the correlations that appear in more months as "main correlation", to

clearly show the relationship between snow density and different influencing variables, as shown in Figure 4b. In general, the correlations between snow density and all influencing variables are very weak, with the maximum average correlation coefficient of only 0.123, which indicates the great difficulty for the estimation task of snow density.

For the 8 snow variables, SD shows apparently higher importance because it has the larger average correlation coefficient of 0.087, followed by ES and SMLT with average correlation coefficient of 0.082. It is noted that snow density is mainly negatively correlated with SF, SA, and SCD, and positively correlated with other snow variables, indicating that the less new snowfall, more snowmelt, and deeper snow depth tend to have higher snow density. Among the 5 meteorological variables, TP has the highest average correlation coefficient of 0.110, indicating that higher precipitation can increase snow density. All five topographical variables show high positive correlation, with average correlation coefficient value of approximately 0.1. Surprisingly, the variable LAI_HV has the largest positive correlation coefficient among all the variables, indicating the importance of vegetation for snow density estimation. In summary, LAI_HV has the strongest correlation with snow density, followed by the TP, SD, and topographic variables among the 20 variables.

[Figure]

Figure 4. Correlation coefficients between snow density and its influencing variables in each month (a), and the average value of the positive and negative correlation coefficients, where the main correlation marked as shade refers to the positive or the negative correlation that occurs in more months than the other (b).

**5. L200, Does the reason for the lower accuracies in Northeast China-Inner Mongolia consider the effect of different underlying surfaces on snow density? Forests and farmland in the Northeast, and grasslands in Inner Mongolia may have very different effects on snow density.**

*Response from Authors:*

The authors greatly agree with this suggestion that the different underlying surface will affect the estimation of snow density. Actually, we have considered the influence of underlying surfaces by using the leaf area index of high vegetation (LAI_HV), which is the one-half of the total green leaf area per unit horizontal ground surface area for high vegetation type. The variable LAI_HV has the largest average correlation coefficient among all the variables, as shown in revised Figure 4.

Moreover, according to the advice, we downloaded the MODIS land vegetation classification product

(MCD12Q1_V6) with a spatial resolution of 500 m and resample them to 25 km to further explore the influence of different land cover on snow density. The correlation relationship between snow density and MCD12Q1 is relatively weak compared with other influencing variables, as shown in Figure 4. We calculated the accuracy of GTWNN model after inputting the MCD12Q1 into the models as an influencing variable, which is slightly improved in comparison with the original GTWNN model without inputting MCD12Q1, with $R^2$ 0.531, RMSE 0.043 g/cm$^3$, and MAE 0.028 g/cm$^3$, as shown in Table S2. In addition, it is noted that the accuracy improvement by inputting MCD12Q1 in Northeast China-Inner Mongolia is higher than that in Xinjiang and Tibetan Plateau.

Therefore, both the high correlation value of the LAI_HV variable and the improvement by inputting MCD12Q1 demonstrate the importance of underlying surfaces for estimating snow density, especially for the Northeast China-Inner Mongolia. Accordingly, the lower accuracies in Northeast China-Inner Mongolia would mainly be caused by the more forest and less stable snow cover in Northeast China-Inner Mongolia than in Xinjiang (See revised section 4.2.2 Accuracies of GTWNN Model in Different Regions), and the MCD12Q1 variable is input in GTWNN model. Hope our efforts have addressed the major concerns.

**Table S2. Impact of MCD12Q1 on accuracies of estimated snow density.**

| Model | Region | Slope | $R^2$ | RMSE | MAE |
|---|---|---|---|---|---|
| GTWNN model without the input of MCD12Q1 | China | 0.975 | 0.524 | 0.043 | 0.028 |
| | Xinjiang | 1.004 | 0.642 | 0.037 | 0.024 |
| | Northeast China-Inner Mongolia | 1.014 | 0.569 | 0.040 | 0.024 |
| | Tibetan Plateau | 0.907 | 0.528 | 0.054 | 0.039 |
| | Other areas | 0.770 | 0.190 | 0.054 | 0.038 |
| GTWNN model with the input of MCD12Q1 | China | 0.906 | 0.531 | 0.043 | 0.028 |
| | Xinjiang | 0.962 | 0.633 | 0.038 | 0.022 |
| | Northeast China-Inner Mongolia | 0.938 | 0.597 | 0.038 | 0.024 |
| | Tibetan Plateau | 0.893 | 0.517 | 0.054 | 0.039 |
| | Other areas | 0.639 | 0.183 | 0.055 | 0.039 |

**6. The reasons for the slightly lower accuracy in the snow melting and accumulation periods are not only the rapid changes in the snow density itself, and insufficient sampling in observation time and space, but also because the snow accumulation in the early stage of snow accumulation is less, and the water content when the snow melts Therefore, the observation is more difficult, and the observation error is relatively large.**

*Response from Authors:*

Thanks for providing the professional suggestion. These comments consider the snow characteristics during the snow accumulation and melt periods, which lead to increased observation difficulty and measurement error, and thus can more comprehensively explain the reason for the slightly lower accuracy in these periods. We added the reason in our revised manuscript.

*4.2.3 Accuracies of GTWNN Model in Different Months*

It is noted that the observation error cannot be ignored, which may be caused by the less snow in the early stage of snow accumulation period, or the large water content when the snowmelt period, making the observation more difficult.

*5.1 Essential Issues of Constructing and Applying GTWNN Model*

In contrast, if the snow changes rapidly, distributes sparsely, or the observation difficulty increases, such as in the early snow accumulation period and the late snowmelt period, the estimated snow density would be less credible and need to be used with caution.

**7. The verification result of the snow density of ERA5 is worse than that of the model in this paper, but many**

**parameters of ERA5 are used in the machine learning model of this paper, so the accuracy of these parameters, if there is also a large error, will not affect the final model accuracy?**

*Response from Authors:*

Thanks for this comment. The reason for choosing the ERA-5 data is that the high spatiotemporal resolution and rich variables compared to other reanalysis data, with a spatial resolution of 0.1° and a temporal resolution of one hour. The near-surface meteorological state and flux fields, including the air temperature, wind speed, surface pressure, and total precipitation, are corrected for the altitude differences and have improved quality (Muñoz-Sabater et al., 2021). The reanalysis can provide an estimation of the meteorological gridded dataset by assimilating various observations into the forecast model system (Dee et al., 2014). The ECMWF ERA-5 land hourly dataset, as any other simulation, provides estimates which have some degree of uncertainty.

To verify whether the accuracy of the influencing variables affect the final model accuracy, we downloaded the instantaneous near surface (2 m) air temperature and precipitation from the China meteorological forcing dataset (CMFD), with a spatial resolution of 0.1° for comparison. CMFD is the high spatial-temporal resolution gridded near-surface meteorological dataset in China, which was made through fusing remote sensing products, reanalysis datasets, and in-situ station data (He et al., 2020). Since CMFD only provides data until 2018, we use CMFD data to replace the temperature and precipitation data of ERA-5, and the accuracies of the models with different influencing variables from 2013 to 2017 are shown in Table 3. The accuracies of new model with CMFD are slightly higher than those of original model indicated by $R^2$, but most RMSE and MAE remain the same.

According to the above results, we can conclude that the accuracy of influencing variables would affect the final model accuracy. Even though the accuracy of ERA-5 snow density is worse than ours, the temperature and precipitation data of ERA-5 achieve comparable performance with CMFD data for driving our model. In addition, considering the high spatiotemporal resolution and rich variables, especially the temporal coverage of ERA-5 data (1950–), we finally choose the ERA-5 data in this study.

We added discussion about the impact of low accuracy of influencing variables on snow density estimation in the revised manuscript. Hope our efforts have addressed the major concerns.

*5.1 Essential Issues of Constructing and Applying GTWNN Model*

In addition, the accuracy of influencing variables would also affect the GTWNN model estimation accuracy. We downloaded the instantaneous near surface (2 m) air temperature and precipitation from the China meteorological forcing dataset (CMFD), with a spatial resolution of 0.1° for comparison. CMFD is the high spatial-temporal resolution gridded near-surface meteorological dataset in China, which was made through fusing remote sensing products, reanalysis datasets, and in-situ station data (He et al., 2020). Since CMFD only provides data until 2018, we use CMFD data to replace the temperature and precipitation data of ERA-5, and the accuracies of the models with different influencing variables from 2013 to 2017 are shown in Table 3. The accuracies of the new model with CMFD are slightly higher than those of original model indicated by R2, but the RMSE and MAE remain the same. However, considering the high spatiotemporal resolution and rich variables, especially the temporal coverage of ERA-5 data (1950–), we finally choose the ERA-5 data in our study.

**Table 3. Accuracy comparison of estimated snow density with different sources of influencing variables.**

| Year | GTWNN model with ERA-5 data | | | GTWNN model with CMFD data | | |
|------|-------|-------|-------|-------|-------|-------|
|      | $R^2$ | RMSE | MAE | $R^2$ | RMSE | MAE |
| 2013 | 0.499 | 0.041 | 0.025 | 0.495 | 0.041 | 0.025 |
| 2014 | 0.521 | 0.040 | 0.024 | 0.531 | 0.039 | 0.024 |
| 2015 | 0.473 | 0.042 | 0.027 | 0.482 | 0.042 | 0.027 |
| 2016 | 0.560 | 0.038 | 0.023 | 0.575 | 0.037 | 0.023 |
| 2017 | 0.591 | 0.040 | 0.023 | 0.587 | 0.040 | 0.023 |
| Overall | 0.529 | 0.040 | 0.024 | 0.534 | 0.040 | 0.024 |

**References**

190    Dee, D. P., Balmaseda, M., Balsamo, G., Engelen, R., Simmons, A. J., and Thépaut, J.-N.: Toward a Consistent Reanalysis of the Climate System. Bulletin of the American Meteorological Society, 95(8), 1235–1248, https://doi.org/10.1175/BAMS-D-13-00043.1, 2014.

He, J., Yang, K., Tang, W., Lu, H., Qin, J., Chen, Y. and Li, X.: The first high-resolution meteorological forcing dataset for land process studies over China, Scientific Data, 7, 25, https://doi.org/10.6084/m9.figshare.11558439, 2020.

Jonas, T., Marty, C., and Magnusson, J.: Estimating the snow water equivalent from snow depth measurements in the Swiss Alps, Journal of

195    Hydrology, 378, 161–167, https://doi.org/10.1016/j.jhydrol.2009.09.021, 2009.

Muñoz-Sabater, J., Dutra, E., Agustí-Panareda, A., Albergel, C., Arduini, G., Balsamo, G., Boussetta, S., Choulga, M., Harrigan, S., Hersbach, H., Martens, B., Miralles, D.G., Piles, M., Rodríguez-Fernández, N.J., Zsoter, E., Buontempo, C., Thépaut, J.-N.: ERA5-Land: a state-of-the-art global reanalysis dataset for land applications, Earth System Science Data, 13, 4349–4383. https://doi.org/10.5194/essd-13-4349-2021, 2021.

200    Sturm, M., Taras, B., Liston, G. E., Derksen, C., Jonas, T., and Lea, J.: Estimating snow water equivalent using snow depth data and climate classes, Journal of Hydrometeorology., 11, 1380–1394, https://doi.org/10.1175/2010JHM1202.1, 2010

**Our authors greatly appreciate the advices and suggestions, and we tried to further improve our work accordingly, hope the revisions could successfully address the raised concerns. Many thanks again!**

205

**This work employed geographically and temporally weighted neural network (GTWNN) model to construct the daily snow density grid products in China, with the support of satellite, ground, and reanalysis data, which is useful for estimating water resources and predicting natural disasters. However, some**
210 **important issues need to be addressed. The details are as follows.**

**1.In terms of abstract, the content needs to be specified and well organized. For instance, in Line 6 of this section, the detailed results supporting that the GTWNN model can improve the estimation of snow density should be given; and in Line 10, the specific models should be listed.**

*Response from Authors:*

215 Thank you very much for this comment. According to the advice, we revised the abstract accordingly and added the detailed results comparing with other regression models and snow density product. The revised Abstract is as follows.

*Abstract.*

Snow density plays a critical role in estimating water resources and predicting natural disasters such as floods,
220 avalanches, and snowstorms. However, gridded products for snow density are lacking for understanding its spatiotemporal patterns. In this study, considering the strong spatiotemporal heterogeneity of snow density, as well as the weak and nonlinear relationship between snow density and the meteorological, topographic, vegetation, and snow variables, the geographically and temporally weighted neural network (GTWNN) model is constructed for estimating daily snow density in China from 2013 to 2020, with the support of satellite, ground, and reanalysis data.
225 The leaf area index of high vegetation, total precipitation, snow depth, and topographic variables are found to be closely related to snow density among the 20 influencing variables. The 10-fold cross-validation results show that the GTWNN model achieves the $R^2$ of 0.531 and RMSE of 0.043 $g/cm^3$, outperforming the geographically and temporally weighted regression model ($R^2 = 0.271$), geographically weighted neural network model ($R^2 = 0.124$), and reanalysis snow density product ($R^2 = 0.095$), which demonstrates the superiority of the GTWNN model by
230 capturing the spatiotemporal heterogeneity of snow density and the nonlinear relationship to the influencing variables. The performance of the GTWNN model is closely related to the state and amount of snow, in which more stable and plentiful snow would result in higher snow density estimation accuracy. With the benefit of the daily snow density map, we are able to obtain knowledge of the spatiotemporal pattern and heterogeneity of snow density in China. The proposed GTWNN model holds the potential for large-scale daily snow density mapping, which will
235 be beneficial for snow parameter estimation and water resource management.

**2.In the introduction, it is not clear why satellite, ground, and reanalysis data are used.**

*Response from Authors:*

Thanks for your question. To achieve daily snow density mapping and understand the relationship between snow density and its influencing variables, we input the multi-source data into the GTWNN model, including the
240 satellite, ground, and reanalysis data. Among the three kinds of data, ground observation data has high accuracy but limited numbers because of the sparsity of stations, serving as the true value of snow density estimation. Satellite data is used to provide information of the snow-related influencing variables, and reanalysis data is used to provide information of the meteorology-related influencing variables for estimating snow density.

We revised the Introduction (seventh paragraph) to illustrate the reasons why satellite, ground, and reanalysis
245 data are used, and the revisions are shown as below.

*1 Introduction*

Geographically weighted regression (GWR) is a model that considers spatial heterogeneity by using local multiple linear regression technology (Fotheringham et al., 1998). To further incorporate temporal dependency, geographically and temporally weighted regression (GTWR) model has been introduced for many disciplines, such as meteorology, hydrology, and social economics (Chen et al., 2017; He and Huang, 2018; Huang et al., 2010). The machine learning approaches such as Random Forests (RF) (Breiman, 2001) and General Regression Neural Network (GRNN) (Specht, 1991) have become popular to fit nonlinear relationships, and it is in the initial stage for estimating snow density (Broxton et al., 2019). We can incorporate geographical and temporal weights into a neural network model to capture the spatiotemporally various and nonlinear relationship between snow density and its influencing variables. In addition, considering the impact of different influencing variables, the satellite data can provide information of the snow-related and topography-related variables, and the reanalysis data can provide information of the meteorology-related variables for estimating snow density based on the true value provided by ground observations. Consequently, to achieve large-scale snow density mapping, we can develop a geographically and temporally weighted neural network (GTWNN) model by considering the multiple influencing variables with the support of satellite, ground, and reanalysis data, which not only considers the spatiotemporal heterogeneity for snow density, but also explains the nonlinear relationship between snow density and different influencing variables.

**3. In the end of Section 2.1, the snow season is divided into three periods. Considering the climate and environment show great spatial heterogeneity in snow cover areas in China, this division of snow season should be expounded.**

*Response from Authors:*

Thanks for the constructive suggestion. The division of different snow periods is only used for analyzing the snow density estimation results rather than for estimating snow density. We greatly agree with that the different snow cover regions show great spatial heterogeneity in terms of climate and environment in China. For example, Xinjiang has less precipitation than Northeast China-Inner Mongolia, and the environment and climate in Tibetan Plateau are distinct due to the high elevation. In addition, the snow density is affected by many influencing variables, such as meteorological variables (e.g., temperature, wind), snow variables (e.g., snow depth), topographic variables (e.g., elevation, slope), and latitude, etc., which are also various among different snow cover regions. However, to ensure the results analysis being comparable, the division of snow periods should be the same in different snow cover regions. Accordingly, we roughly divide the different snow periods according to the seasons in China.

Specifically, the hydrological year is defined as the time period from 1 September to 31 August of the next year in China, in which September, October, and November are treated as the autumn season, December, January, and February as the winter season, March, April, and May as the spring season (Ke et al., 2016; Sun et al., 2020). The three snow periods (snow accumulation period, snow stable period, and snowmelt period) are divided according to the autumn, winter, and spring seasons.

We added the reason for the division of snow season in the revised manuscript, which is shown as follows.

*2.1 In Situ Snow Density*

Therefore, the study focuses on estimating snow density in the snow season from September to May of the next year. To further analyze the estimation results, the snow season is roughly divided into the snow accumulation period (September –November, autumn), the snow stable period (December–February of the next year, winter), and the snowmelt period (March–May, spring) according to the division of season (Ke et al., 2016).

**4. In terms of Equation 2, not all the variables are explained in detail.**

*Response from Authors:*

Thanks for the comment on details. We swapped the position of Equation 2 and Equation 3, and corrected the problems you pointed out, which makes the introduction of GTWNN model more reasonable. The revisions are shown as below.

**3.1 GTWNN Model**

$$d_{ST}^i = \sqrt{\left[\left(j_p - j_s^i\right)^2 + \left(k_p - k_s^i\right)^2\right] + \varphi(t_p - t_s^i)^2} , \tag{2}$$

$$W_{GT}^i = \begin{cases} [1 - \left(\frac{d_{ST}^i}{h_{ST}}\right)^2]^2, & d_{ST}^i < h_{ST} \\ 0, & d_{ST}^i \geq h_{ST} \end{cases} , \tag{3}$$

295    where $d_{ST}^i$ denotes the spatial and temporal distance between the $i_{th}$ ($i$=1, 2, …, $N$) sample point ($s$) and the prediction point ($p$), in which $j$ and $k$ represent the location of point; and $t$ represents the time, as shown in Figure 2. $W_{GT}^i$ indicates the weight of the sample point in the GTW model.

**5.The variables in Figure 2 should be explained in or below this picture.**

*Response from Authors:*

300    Thanks for your careful review. According to the suggestion, we revised Figure 2 by explaining all the important variables for each module. The revised Figure 2 is shown as below. Hope our efforts have addressed the concern.

[Figure]

**Figure 2. Schematic of the GTWNN model for the estimation of snow density, where GTW refers to the geographically**
305    **and temporally weighted, and GRNN refers to the generalized regression neural network.**

**6.How each kinds of data are used specifically is not given in Section 2 or 3.**

*Response from Authors:*

Thank you very much for this question. Totally, three kinds of data are used in our study, including ground observation data, satellite data, and reanalysis data, respectively. As answered in Question 2, the ground observation
310    data serves as the true value of snow density. The satellite data is used to provide information of the snow-related influencing variables, and the reanalysis data is used to provide information of the meteorology-related influencing variables for estimating snow density.

To clarify how the different data are used, we rewrote Section 2.3 Data Integration to clarify how to preprocess and to use each kind of data. In addition, we revised Section 3.1 to further explain the role of each data in the
315    GTWNN model. The revisions are shown as below.

**2.3 Data Integration**

Three kinds of data are used, including ground observation data, satellite data, and reanalysis data, where the ground observation data is used to provide the true value of snow density, and the satellite and reanalysis data are used to provide information of different influencing variables of snow density. Before the model development, data

pre-processing is conducted. Firstly, since the spatial resolution varies among the different satellite data and reanalysis data, they are resampled to 25 km for snow density mapping using average or accumulation methods depending on the data type. The spatial resolution of 25 km is determined to match that of most SD and SWE products by passive microwave remote sensing. The elevation and slope are resampled to 25 km by average, and the standard deviation of elevation (ELEVATION_STD) and slope (SLOPE_STD) are also calculated to reflect the topographic relief within the range of 25 km. Accordingly, the ground observations of snow density measured at multiple sites are averaged for each 25 km grid cell. In addition, the min-max normalization method is applied to normalize different influencing variables. After that, we collect 16935 samples for model establishment and validation, where a sample refers to a grid cell with ground observations of snow density and its influencing variables.

**3.1 GTWNN Model**

The GTWNN model is a spatiotemporally aware model composed of a geographically and temporally weighted (GTW) model to capture spatiotemporal heterogeneity and a generalized regression neural network (GRNN) to deal with the weak and nonlinear relationships between snow density and its influencing variables, including the meteorological variables, topographical variables, vegetation variables, and snow variables, which could be expressed as shown in Eq. (1), and its schematic is shown in Figure 2.

$$snow\ density = f_{(S,T)}(x, y) \ , \tag{1}$$

where $snow\ density$ is the estimated snow density in each cell; $(S, T)$ presents the spatial and temporal distance between the sample point and the prediction point; $x$ refers to the influencing variables of snow density; and $y$ refers to the ground observation data.

**7.In Section 3.2, the title of this section is not appropriate for the content. In addition, how to evaluate the GTWNN model (such as the metrics to evaluate the performance) should be described.**

*Response from Authors:*

Thanks for this constructive suggestion. We revised Section 3.2 and added the metrics to evaluate the model performance. We agree with that the title of Section 3.2 (Model Evaluation) is not appropriate for the content, because in addition to the validation methods (such as the 10-fold cross-validation technique and metrics), we also described methods for selecting the parameters of the GTWNN model. Hence, we modified the title of this section to be 3.2 Parameter Determination and Model Evaluation Method. The revisions are presented as below. Hope our efforts have addressed the major concerns.

**3.2 Parameter Determination and Model Evaluation Method**

There are three essential parameters in GTWNN model, including the spatiotemporal bandwidth $h_{ST}$ and the scale factor $\varphi$ of the GTW model, and the $spread$ of GRNN model. To evaluate the model performance as well as to determine the optimal parameters, the 10-fold cross-validation technique is adopted (Fotheringham et al., 2003; Rodriguez et al., 2010); that is, all the collected samples are randomly divided into 10 folds, nine folds are exploited for the model fitting, and one fold is used for the validation. The above steps are repeated 10 times so as to evaluate the model performance on each fold of the validation samples. Finally, a scale factor $\varphi$ of 0.01, $spread$ of 0.5, and an adaptive bandwidth regime $h_{ST}$ of 8 are obtained, which can achieve the best performance.

In addition, the coefficient of determination ($R^2$, unitless), the mean absolute prediction error (MAE, g/cm$^3$), and the root mean squared prediction error (RMSE, g/cm$^3$) are adopted to evaluate the performance of the GTWNN model.

**8.In table 2, the details about each model are deficient.**

*Response from Authors:*

The authors greatly thank for the comment. The aim of comparison with other regression models is to demonstrate the superiority of the GTWNN model on capturing the spatiotemporal heterogeneity of snow density

and nonlinear relationship between influencing variables and snow density. We added the details about each model for comparison, including the differences among 6 models and the corresponding comparison results (See revised 4.3.1 Comparison with Other Regression Models). In addition, we revised the caption of Table 2 to clarify the meaning of each model. The revisions are shown as below.

**4.3.1 Comparison with Other Regression Models**

The GTWNN model is compared with five other regression models to demonstrate its advantages for snow density estimation by capturing the spatiotemporal heterogeneity of snow density and its nonlinear relationship to influencing variables, as shown in Table 2. The models involved for comparison include the multiple linear regression (R) model, geographically weighted regression (GWR) model, geographically and temporally weighted regression (GTWR) model, general regression neural network (GRNN) model, and geographically weighted neural network (GWNN) model. It is noted that the original R and GRNN models are global regression models established on all samples, regardless of the geographical and temporal weights. The R model captures the linear relationship between snow density and its influencing variables, and the GRNN has nonlinear mapping ability (Specht, 1991). Meanwhile, the GWR (Fotheringham et al., 1998) and GWNN models are spatial local models constructed from R and GRNN by setting a bandwidth $h$, in which the sample points have different weights ($W_G$) according to the spatial distance ($d_s$). The GTWR and GTWNN models further incorporate temporal dependencies, which adds a new scale factor $\varphi$ to balance the different weights of the spatial and temporal distances. The optimal parameters of the compared models are determined by the 10-fold cross validation strategy as that for the GTWNN model.

Table 2. Accuracies of various regression models for estimating daily snow density.

| Model | Full name of model | Slope | $R^2$ | RMSE | MAE |
|-------|--------------------|-------|-------|------|-----|
| R | Multiple linear regression | 0.783 | 0.015 | 0.060 | 0.044 |
| GWR | Geographically weighted regression | 0.069 | 0.022 | 0.143 | 0.091 |
| GTWR | Geographically and temporally weighted regression | 0.398 | 0.271 | 0.070 | 0.043 |
| GRNN | General regression neural network | 2.394 | 0.033 | 0.062 | 0.046 |
| GWNN | Geographically weighted neural network | 0.489 | 0.124 | 0.062 | 0.043 |
| GTWNN | Geographically and temporally weighted neural network | 0.906 | 0.531 | 0.043 | 0.028 |

**9.The text and logic in the manuscript needs improved, particularly in the Results Section. For example, what's the relationship between Section 4.1 (Descriptive Statistics of Ground Observations) and other results? The position of Section 4.2.3 (Importance of the Influencing Variables for Snow Density Estimation) needs consideration.**

*Response from Authors:*

Thanks for your comments. According to the suggestions, we reviewed all the text and logic in the manuscript and made revisions accordingly, especially in the Results Section.

Firstly, in Section 4.1 Descriptive Statistics of Ground Observations, the aim of the descriptive statistics of the snow density in different years, months, and snow cover regions is to show the spatiotemporal distribution of snow density in China, based on the observed values at the stations, which can be used to verify the results of snow density mapping (as described in 4.4 Mapping of Snow Density). In addition, the snow density box plots (as shown in Figure 3), including the mean, median, upper and lower quartile value (25%~75%), and upper and lower whisker range (1%~99%), show the dispersion and variation fluctuations in snow density, together with the number of ground observations in different times and areas, which may be the reasons for the accuracy performance of the GTWNN model estimation in different snow cover regions and months (as describe in 4.2.2 Accuracies of GTWNN Model in Different Regions). We add the aim of Section 4.1 and the relationships to other results, as below.

**4.1 Descriptive Statistics of Ground Observations**

Snow density has strong spatiotemporal heterogeneity, and we calculated statistics of the 16935 samples

generated from ground observations in terms of the snow density and the number of observations in different years, months, and snow cover regions, as shown in Figure 3, which show the dispersion and variation fluctuations in snow density and can be used to verify the results of snow density mapping.

**4.2.2 Accuracies of GTWNN Model in Different Regions**

The Tibetan Plateau has the lowest $R^2$ of 0.517 and the highest RMSE and MAE, which is mainly caused by the high variation fluctuations of snow density and sparse meteorological stations, as indicated in Figure 3c.

**4.4.2 Temporal Change**

In addition, the monthly mean snow density from the estimated daily snow density map in Figure 9f shows a similar pattern with that from the ground observations in Figure 3b, which further demonstrates the effectiveness of the proposed GTWNN model for snow density estimation.

Secondly, we reorganized the content and logic of the Result Section and adjusted the position of original Section 4.2.3, and now the content of Section 4.2 includes 4.2.1 Relationship between Snow Density and Its Influencing Variables, 4.2.2 Accuracies of GTWNN Model in Different Regions, and 4.2.3 Accuracies of GTWNN Model in Different Months.

As a whole, in Section 4, the first part introduces the descriptive statistics of ground observations, which is the basis to explore the spatiotemporal distribution snow density in China based on the ground observations. The second part is model validation, which is presented in order from input influencing variables to output results in different snow cover regions and snow periods. Next, to highlight the superiority of the GTWNN model, the third part is model comparison, which includes the comparison with other regression models and reanalysis snow density product. Finally, the estimated results are mapped to reveal the spatiotemporal distribution of now density.

**10. Snow density and its CV in different snow cover regions vary apparently, as well as the monthly changes of snow density. While, the explications about these phenomena are limited in the present manuscript.**

*Response from Authors:*

Thank you very much for providing the professional suggestion. We added the content about the mean snow density and CV in different snow cover regions and periods in Figure 8, and briefly explained the reasons for these phenomena in Section 4.4. The revisions are presented as below.

[revised manuscript text omitted]

**Our authors greatly appreciate the advices and suggestions, and we tried to further improve our work accordingly, hope the revisions could successfully address the raised concerns. Many thanks again!**

**Comments & Suggestions by Referee #2**

**Snow density plays a critical role in the estimation of snow water equivalent (SWE). Predicting a temporally and spatially variant snow density is not trivial and is usually assume constant for SWE estimates. This study presents a geographically and temporally weighted neural network (GTWNN) model to predict daily snow density across China. This work relies on empirical relations with influencing variables and machine learning algorithm, to predict density over time and space. This work proposes a great way to map snow density over China, but further clarifications are needed before publications.**

**In general, no physical understanding of snow density with influencing variable was explored or used in the modelling. This method relies purely on empirical relations. Not those empirical relations cannot be used but perhaps adding a bit more physical understanding in the variables selection or using a physical model at the regional scale could improve this work.**

*Response from Authors:*

Thank you very much for the suggestion. The suggestion about adding physical understanding in the variables selection or using a physical model at the regional scale will be responded in detail in specific comment 9.

**Specific comments:**

**1. L52-56 This paragraph needs more on how topography and vegetation influence snow density. It might also be also useful to define the scale at which these processes operate relative to this work.**

*Response from Authors:*

Thank you very much for this comment. Accordingly, we revised the Introduction part on how topography and vegetation influence snow density in detail, and the revisions are shown as below.

*1 Introduction*

The terrain and surface types also play an important role in snow density (Clark et al., 2011; Judson and Doesken, 2000). For example, snow density was found to be lower at higher elevations, and even decreased by approximately 0.006 g/cm$^3$ with each 100 m increase in elevation (Zhong et al., 2014), which is indirectly affected by energy balance, temperature decreases with elevation in general (Elder et al., 1998). The indirect effect of slope on snow density includes two ways, one is redistribution of snow via avalanching and wind transport, and another is the amount of radiation received, which results in changes in snow grain size, porosity, and density. In addition, the aspect also affects the snow density through radiation, because sunny-facing slopes that experience high radiation inputs will be more likely to have snow melt, introducing liquid water into the snow, which also increase snow density by filling the pore space with liquid water (Wetlaufer et al., 2016). The average snow density in forest areas was 8%–13% less than that in open areas (Zhong et al., 2014), and these observed density differences are attributed to either mass, delivery, wind, or radiation effects (Bonner et al., 2022). Mass effect is a reduction in the snow mass due to canopy interception loss, with lower compaction rates and snow density. Delivery effect refers to that snow is trapped by the canopy and then delivered to the underlying snowpack, either as unloaded snow or draining melt water. Wind effect occurs when wind speed is reduced by forest obstruction, resulting in a higher snow density relative to open areas because of wind packing. Radiation effect can control snow layer temperature, and melt-refreeze cycles to change snow density (Essery et al., 2008; Storck et al., 2002; Winstral and Marks, 2014).

**2. L71 This is true but maybe used them at the regional scale to add a physical basis of energy exchange in the snowpack.**

*Response from Authors:*

Thanks for your question. We agree with that. To make it more rigorous, we revised this sentence as follows.

*1 Introduction*

One method to explain the spatial and temporal variations in snow density is to use a physical model, such as the coupled energy and mass-balance model ISNOBAL (Hedrick et al., 2018; Marks et al., 1999), which can explicitly simulate a number of snowpack properties including snow density and SWE at the regional scale, and add a physical basis of energy exchange in the snowpack. However, snow density physical models are complex and cannot achieve large-scale spatialization of snow density (Raleigh and Small, 2017).

**3. L120 More is needed here on how the topographic parameters were estimated. Was the mean of all pixels at 30m resolution used to estimate the elevation? Could the standard deviation or other statistical parameters of sub pixel variability be used?**

*Response from Authors:*

Thanks for the comment in detail. Accordingly, we added the details on how to estimate the topographic parameters. The slope and aspect are firstly calculated by elevation at 30 m resolution, and they are resampled to 25 km to get the mean values. In addition, according to the suggestion, we tried to calculate the standard deviation of elevation and slope (ELEVATION_STD and SLOPE_STD), which reflect the topographic relief within the range of 25 km, and to explore the effectiveness of the two statistical parameters for snow density estimation. The performance of the new GTWNN model with additional input of ELEVATION_STD and SLOPE_STD are shown in Table S1 as below.

As shown in Table S1, the $R^2$ of new models are higher than that of original model in 5 of 9 years, and the overall accuracy is slightly improved from 0.515 to 0.516. Although the standard deviation of elevation and slope cannot apparently improve the GTWNN model accuracy, it is still an influencing variable closely related to the snow density. Therefore, we added the ELEVATION_STD and SLOPE_STD in our new model and updated all the related results in the revised manuscript.

**Table S1. Accuracy of estimated snow density with different influencing variables.**

| Year | Original model | | | New model | | |
|---|---|---|---|---|---|---|
| | $R^2$ | RMSE | MAE | $R^2$ | RMSE | MAE |
| 2013 | 0.484 | 0.041 | 0.027 | 0.485 | 0.041 | 0.027 |
| 2014 | 0.470 | 0.041 | 0.026 | 0.504 | 0.040 | 0.026 |
| 2015 | 0.439 | 0.043 | 0.029 | 0.466 | 0.042 | 0.028 |
| 2016 | 0.526 | 0.039 | 0.025 | 0.516 | 0.040 | 0.026 |
| 2017 | 0.526 | 0.043 | 0.027 | 0.528 | 0.042 | 0.027 |
| 2018 | 0.518 | 0.050 | 0.033 | 0.493 | 0.051 | 0.035 |
| 2019 | 0.620 | 0.049 | 0.033 | 0.606 | 0.050 | 0.033 |
| 2020 | 0.508 | 0.045 | 0.026 | 0.525 | 0.044 | 0.025 |
| Overall | 0.515 | 0.043 | 0.028 | 0.516 | 0.043 | 0.028 |

The revisions are shown as below.

*2.2 Satellite and Reanalysis Data*

The topographical variables of elevation are obtained from the Shuttle Radar Topography Mission (SRTM) digital elevation model with a spatial resolution of 30 m, and then slope and aspect are derived based on the elevation.

*2.3 Data Integration*

Before the model development, data pre-processing is conducted. Firstly, since the spatial resolution varies among the different satellite data and reanalysis data, they are resampled to 25 km for snow density mapping using average or accumulation resampling methods depending on the data type. The spatial resolution of 25 km is determined to match that of most SD and SWE products by passive microwave remote sensing. The elevation and

slope are resampled to 25 km by average, and the standard deviation of elevation (ELEVATION_STD) and slope (SLOPE_STD) are also calculated to reflect the topographic relief within the range of 25 km.

**4. Section 3.2 It is not clear how the model is evaluated… against ground observations? It says in the objectives that daily snow density mapping is achieve by integrating satellite, ground and reanalysis data. One or two sentences are needed here to clarify which is used for what and how the model is trained and validated.**

*Response from Authors:*

Thanks for your careful review for our manuscript. Firstly, we revised the title of Section 3.2 to Parameter Selection and Model Evaluation Method, in which the metrics to evaluate the model performance is added, and the methods for selecting the optimal parameters of the GTWNN model and validation methods also are revised in detail.

Secondly, we agree with that some sentences about how each kinds of data are used and how the model is trained and validated need to be added. Generally, we input the multi-source data into the GTWNN model, including the satellite, ground and reanalysis data. Among the three kinds of data, ground observation data has high accuracy but limited numbers because of the sparsity of stations, serving as the true value of snow density. Satellite data is used to provide information of the snow-related influencing variables, and reanalysis data is used to provide information of the meteorology-related influencing variables for estimating snow density. We finally collect 16935 samples after data preprocessing, where a sample refers to a grid cell with ground observations of snow density and its influencing variables. The 10-fold cross-validation technique is used to evaluate the model performance and determine the optimal parameters, that is, all the collected samples are randomly divided into 10 folds, in which nine folds are exploited for the model fitting, and one fold is used for the validation. The above steps are repeated 10 times so as to evaluate the model performance on each fold of the validation samples, therefore, all samples are both training and validation dataset but in different folds.

Accordingly, we revised Section 2.3 to illustrate how each kind of data is used in this study, and also revised Section 3.2 to illustrate 10-fold model validation method.

**2.3 Data Integration**

Three kinds of data are used, including ground observation data, satellite data, and reanalysis data, where the ground observation data is used to provide the true value of snow density, and the satellite and reanalysis data are used to provide information of different influencing variables of snow density.

**3.2 Parameter Determination and Model Evaluation Method**

There are three essential parameters in GTWNN model, including the spatiotemporal bandwidth $h_{ST}$ and the scale factor $\varphi$ of the GTW model, and the *spread* of GRNN model. To evaluate the model performance as well as to determine the optimal parameters, the 10-fold cross-validation technique is adopted (Fotheringham et al., 2003; Rodriguez et al., 2010); that is, the collected samples are randomly divided into 10 folds, nine folds are exploited for the model fitting, and one fold is used for the validation. The above steps are repeated 10 times so as to evaluate the model performance on each fold of the validation samples. Finally, a scale factor $\varphi$ of 0.01, *spread* of 0.5, and an adaptive bandwidth regime $h_{ST}$ of 8 are obtained, which can achieve the best performance.

In addition, the coefficient of determination ($R^2$, unitless), the mean absolute prediction error (MAE, g/cm$^3$), and the root mean squared prediction error (RMSE, g/cm$^3$) are adopted to evaluate the performance of the GTWNN model.

**5. Figure 4 Again, how was it trained and validated. Can you define the dataset percentage used for training and validation? Was it trained on some years and evaluated on the remaining years and same for the region?**

*Response from Authors:*

Thank you very much for this question. Originally, we built the models for each year from 2013 to 2020. The

models are trained and validated using the 10-fold cross-validation technique, as answered in Question 4. For each year, all the collected samples are randomly divided into 10 folds, nine folds are exploited for the model fitting, and one fold is used for the validation, and then the above steps are repeated 10 times so as to evaluate the model performance on each fold of the validation samples. Hence, we finally obtain the estimated snow density of all data, which are compared with the observed snow density to validate the model accuracy in different periods and snow cover regions.

In fact, the GTWNN model is a spatiotemporal interpolation model based on the ground observation snow density, and is constructed separately for each year with the consideration of the snow variety in different years, which cannot be trained on some years and evaluated on the remaining years. We revised the introduction of GTWNN Model to clarify this, and the revisions are shown as below.

**5.2 Advantages and Limitations**

It is noted that the GTWNN model is a spatiotemporal interpolation model based on the ground observation snow density, and the confidence of the snow density map produced by the GTWNN model is still constrained by the distribution of the observation stations

**6. Section 4.2.3 Other methods than Pearson correlation factor can be used to investigate the importance of influencing variables. This only indicates a correlation. I would suggest using a permutation importance-based method or an impurity importance from a tree classifier. Maybe it would give better insight on the variables.**

*Response from Authors:*

Thanks for this constructive suggestion. Accordingly, we tried the permutation importance-based method to investigate the importance of influencing variables. According to our understanding, the process of permutation importance-based method is as follow: (1) put all influencing variables into the GTWNN model and get the baseline metric, defined by $R^2$; (2) scramble each variable column in turn, and input into the model to evaluate the metric again; (3) calculate the differences between the baseline metric and metric by permutating the variables column, reflecting the importance of different influencing variables. However, since the influencing variables are weak for snow density, the results of the scrambled influencing variables are random, and the corresponding 10-fold cross-validation results are also random. Hence, we repeat the permutation importance-based method many times and filter out different influencing variables. Finally, we calculate the mean of differences between the baseline metric and metric by permutating the different variables column, and the results are shown in Figure S1.

Compared with original Figure 4, more influencing variables are selected by the permutation importance-based method in different years than the stepwise regression method, especially after 2016. However, the importance of different influencing variables indicated by the permutation importance-based method is similar with each other, and the most important variables are ES, SF, SCA, SCD. It is noted that the LAI_HV and topographic variables are still important. In addition, we can find that the difference values of different influencing variables are so small and most of the values are below 0.01, which indicates that the input of GTWNN model has little effect on the model accuracy.

[Figure]

**Original Figure 4. Correlation coefficient between snow density and its influencing variables selected by the stepwise regression method in each year (a), and the average of the absolute value of the correlation coefficient and the number of selections within these years (b).**

[Figure]

**Figure S1. Differences between the baseline metric and metric by permutating the different influencing variables column in each year (a), and the average of the absolute value of the correlation coefficient and the number of selections within these years (b).**

We further compare the effect of different variable selection methods on the performance of GTWNN model. The selected influencing variables involved for comparison include all the variables, variables selected by the permutation importance-based method, variables with the top 30% selected by the permutation importance-based

method, and variables selected by the stepwise regression method, and the model accuracies are shown in Table S2. The permutation importance-based method is effective for improving the $R^2$ of the GTWNN model, in comparison with the stepwise regression method. Surprisingly, if we input all influencing variables into GTWNN model, the $R^2$ is higher than the other models.

**Table S2. Accuracies of various methods for investigate the importance of influencing variables.**

| Influencing variables | Slope | $R^2$ | RMSE | MAE |
|---|---|---|---|---|
| All variables | 0.978 | 0.519 | 0.043 | 0.028 |
| Permutation importance | 0.984 | 0.518 | 0.043 | 0.028 |
| Permutation importance with 30% | 0.997 | 0.512 | 0.043 | 0.028 |
| Stepwise regression | 0.986 | 0.515 | 0.043 | 0.028 |

Finally, we choose to input all influencing variables into the GTWNN model in our revised manuscript, and to calculate the Pearson correlation coefficient between snow density and all the influencing variables in different months rather than in each year, to better understand the relationship between snow density and different influencing variables, as shown in the revised Section 4.2.1 Relationship between Snow Density and Its Influencing Variables. Hope our efforts have addressed the major concerns.

**4.2.1 Relationship between Snow Density and Its Influencing Variables**

The Pearson correlation coefficient between snow density and its influencing variables is calculated to indicate the importance of the variables in each month, as shown in Figure 4a, where September and May are not involved because of the small number of ground observations. The influencing variables and the corresponding correlation coefficient values are various in different months because of the heterogeneity of snow. In addition, we calculate the average value from October to April for the positive and negative correlation coefficients, respectively, to indicate the importance of each influencing variable for snow density. We also count the number of months with positive or negative correlations and mark the correlations that appear in more months as "main correlation", to clearly show the relationship between snow density and different influencing variables, as shown in Figure 4b. In general, the correlations between snow density and all influencing variables are very weak, with the maximum average correlation coefficient of only 0.123, which indicates the great difficulty for the estimation task of snow density.

For the 8 snow variables, SD shows apparently higher importance because it has the larger average correlation coefficient of 0.087, followed by ES and SMLT with average correlation coefficient of 0.082. It is noted that snow density is mainly negatively correlated with SF, SA, and SCD, and positively correlated with other snow variables, indicating that the less new snowfall, more snowmelt, and deeper snow depth tend to have higher snow density. Among the 5 meteorological variables, TP has the highest average correlation coefficient of 0.110, indicating that higher precipitation can increase snow density. All five topographical variables show high positive correlation, with average correlation coefficient value of approximately 0.1. Surprisingly, the variable LAI_HV has the largest positive correlation coefficient among all the variables, indicating the importance of vegetation for snow density estimation. In summary, LAI_HV has the strongest correlation with snow density, followed by the TP, SD, and topographic variables among the 20 variables.

[Figure]

**Figure 4. Correlation coefficients between snow density and its influencing variables in each month (a), and the average value of the positive and negative correlation coefficients, where the main correlation marked as shade refers to the positive or the negative correlation that occurs in more months than the other (b).**

**7. L90 It is stated "to understand how the influencing variables affect snow density estimation". How was this address in the study?**

*Response from Authors:*

Thanks for the constructive suggestion. As answered in Question 6, to understand the relationship between snow density and its influencing variables, we calculated the Pearson correlation coefficient between snow density and all the influencing variables in different months, which is a simple statistical analysis without a physical basis. Therefore, this sentence "to understand how the influencing variables affect snow density estimation" is imprecise, we revised it as "to understand the relationship between snow density and its influencing variables". Hope our efforts have addressed your concern.

*1 Introduction*

to validate the effectiveness of the proposed model in various situations and to understand the relationship between snow density and its influencing variables.

**8. Section 4.3.2 What does this section adds to the manuscript. Does it relate to the objectives? Also, most of the influencing variables come from the ERA-5 reanalysis dataset. Does it affect the results?**

*Response from Authors:*

The authors greatly thank for the comment. According to previous studies, the large-scale daily snow density mapping is currently rare. The reanalysis product ERA-5 can provide the large-scale daily snow density grid dataset, which is produced by comprehensively considering various influencing variables, such as snow pressure, viscosity, near surface air temperature, and wind speed (Muñoz-Sabater, 2019). In addition to highlighting the superiority of the GTWNN model by comparing with other regression models by Section 4.3.1, we want to demonstrate the high accuracy of estimated snow density product by comparing with other large-scale daily snow density products by Section 4.3.2.

In addition, it is noted that the ERA-5 snow density is only used for comparison and not involved in our snow

density estimation model. Many other variables are obtained from ERA-5 for constructing the model. The reason for choosing the ERA-5 data is that the high spatiotemporal resolution and rich variables compared to other reanalysis data, with a spatial resolution of 0.1° and a temporal resolution of one hour. The near-surface meteorological state and flux fields, including the air temperature, wind speed, surface pressure, and total precipitation, are corrected for the altitude differences and have improved quality (Muñoz-Sabater et al., 2021). The reanalysis can provide an estimation of the meteorological gridded dataset by assimilating various observations into the forecast model system (Dee et al., 2014). The ECMWF ERA-5 land hourly dataset, as any other simulation, provides estimates which have some degree of uncertainty.

To verify whether the accuracy of the influencing variables affect the final model accuracy, we downloaded the instantaneous near surface (2 m) air temperature and precipitation from the China meteorological forcing dataset (CMFD), with a spatial resolution of 0.1° for comparison. CMFD is the high spatial-temporal resolution gridded near-surface meteorological dataset in China, which was made through fusing remote sensing products, reanalysis datasets, and in-situ station data (He et al., 2020). Since CMFD only provides data until 2018, we use CMFD data to replace the temperature and precipitation data of ERA-5, and the accuracies of the models with different influencing variables from 2013 to 2017 are shown in Table 3.

The accuracies of new model with CMFD are slightly higher than those of original model indicated by $R^2$, but the RMSE and MAE remain the same, which indicates that the accuracy of the influencing variables will affect the model accuracy, but the temperature and precipitation data of ERA-5 are comparable to that of CMFD for driving our model.

According to the above results, we can conclude that the accuracy of influencing variables would affect the final model accuracy. Even though the accuracy of ERA-5 snow density is worse than ours, the temperature and precipitation data of ERA-5 achieve comparable performance with CMFD data for driving our model. In addition, considering the high spatiotemporal resolution and rich variables, especially the temporal coverage of ERA-5 data (1950–), we finally choose the ERA-5 data in this study.

We added discussion about the impact of low accuracy of influencing variables on snow density estimation in the revised manuscript. Hope our efforts have addressed the major concerns.

**5.1 Essential Issues of Constructing and Applying GTWNN Model**

In addition, the accuracy of influencing variables would also affect the GTWNN model estimation accuracy. We downloaded the instantaneous near surface (2 m) air temperature and precipitation from the China meteorological forcing dataset (CMFD), with a spatial resolution of 0.1° for comparison. CMFD is the high spatial-temporal resolution gridded near-surface meteorological dataset in China, which was made through fusing remote sensing products, reanalysis datasets, and in-situ station data (He et al., 2020). Since CMFD only provides data until 2018, we use CMFD data to replace the temperature and precipitation data of ERA-5, and the accuracies of the models with different influencing variables from 2013 to 2017 are shown in Table 3. The accuracies of the new model with CMFD are slightly higher than those of original model indicated by R2, but the RMSE and MAE remain the same. However, considering the high spatiotemporal resolution and rich variables, especially the temporal coverage of ERA-5 data (1950–), we finally choose the ERA-5 data in our study.

Table 3. Accuracy comparison of estimated snow density with different sources of influencing variables.

| Year | GTWNN model with ERA-5 data | | | GTWNN model with CMFD data | | |
|---|---|---|---|---|---|---|
| | $R^2$ | RMSE | MAE | $R^2$ | RMSE | MAE |
| 2013 | 0.499 | 0.041 | 0.025 | 0.495 | 0.041 | 0.025 |
| 2014 | 0.521 | 0.040 | 0.024 | 0.531 | 0.039 | 0.024 |
| 2015 | 0.473 | 0.042 | 0.027 | 0.482 | 0.042 | 0.027 |
| 2016 | 0.560 | 0.038 | 0.023 | 0.575 | 0.037 | 0.023 |
| 2017 | 0.591 | 0.040 | 0.023 | 0.587 | 0.040 | 0.023 |
| Overall | 0.529 | 0.040 | 0.024 | 0.534 | 0.040 | 0.024 |

**9. Line 363 It is stated that weak correlations exist between snow density and the influencing variables chosen for the predictive model. Could a physical snowpack model (ISNOBAL, CROCUS or SNOWPACK) be used for the 4 different regions (not all pixel) to try to add a physical base to the prediction that is mostly empirical through weak correlations at the moment?**

*Response from Authors:*

Thanks for your professional comments. According to our understanding about the physical snowpack models (ISNOBAL, CROCUS, and SNOWPACK), ISNOBAL (Marks et al., 1999) is a distributed, physically based energy and mass balance snow model that explicitly solves for a number of snowpack properties including snow depth, density, and SWE, CROCUS is the first model to simulate the metamorphism and layering of the snowpack (Brun et al., 1992), which made possible the first real-time distributed simulation of the snowpack over an alpine region for operational avalanche forecasting (Durand et al., 1999), and SNOWPACK is a multi-purpose snow and land-surface model that focuses on a detailed description of the mass and energy exchange between the snow, the atmosphere and optionally with the vegetation cover and the soil. It also includes a detailed treatment of mass and energy fluxes within these media (Lehning et al., 2002a; Lehning et al., 2002b). However, these models were mostly used in a small scale area, the spatial scales of various studies range from 0.015 $km^2$ over a 2.5 m grid (Kormos et al., 2014), 1180 $km^2$ over a 50 m grid (Hedrick et al., 2018), 460 $km^2$ over a 75 m grid (Marks et al., 1999), and 2150 $km^2$ over a 250 m grid (Garen and Marks, 2005), which all have the high spatial resolution at the regional scale. In our study, the estimation task is performed at the resolution of 25 km, which is much coarser than before. Accordingly, we only have the ground observation data about snow parameters, which is sparsely distributed in China. In addition, the meteorological data of the highest spatial resolution is the ERA-5 reanalysis product with the resolution of 0.1°. Even though the snow density physical models can help us understand the relationship between snow density and different influencing variables from a physical mechanism perspective, the above limitations may prevent snow density physical models, even at small regional scales. Hope our efforts have addressed the major concerns.

**10. Line 389 The GTWNN can deal with spatiotemporal heterogeneity but how about temporal and spatial transferability of the model in the training/validation?**

*Response from Authors:*

Thank you very much for the question. The GTWNN model could be simply expressed as $snow\ density = f_{(S,T)}(x, y)$, where $snow\ density$ is the estimated snow density in each cell; $(S, T)$ presents the spatial and temporal distance between the sample point and the prediction point, which is used to select the suitable sample point; and the model input includes $x$ and $y$, $x$ refer to the influencing variable of snow density, $y$ refers to the ground observation data, as shown in Figure 2. As answered in Question 5, the GTWNN model cannot achieve the spatial and temporal transferability. We revised Section 3.1 to further clarify the function of GTWN as below.

*3.1 GTWNN Model*

The GTWNN model is a spatiotemporally aware model composed of a geographically and temporally weighted (GTW) model to capture spatiotemporal heterogeneity and a generalized regression neural network (GRNN) to deal with the weak and nonlinear relationships between snow density and its influencing variables, including the meteorological variables, topographical variables, vegetation variables, and snow variables, which could be expressed as shown in Eq. (1), and its schematic is shown in Figure 2.

$$snow\ density = f_{(S,T)}(x, y)\ , \tag{1}$$

where $snow\ density$ is the estimated snow density in each cell; $(S, T)$ presents the spatial and temporal distance between the sample point and the prediction point; $x$ refers to the influencing variables of snow density; and $y$ refers to the ground observation data.

[Figure]

**Figure 2. Schematic of the GTWNN model for the estimation of snow density, where GTW refers to the geographically and temporally weighted, and GRNN refers to the generalized regression neural network.**

**11. Line 402 How would that be achieved? Using a physical model?**

*Response from Authors:*

Thank you very much for your comment. Since GTWNN model cannot achieve the spatial and temporal transferability, we expect to further develop a snow density prediction model without the dependence of observed snow density for model inference. Intuitively, we think about developing advanced machine learning methods with spatiotemporal awareness, such as the Geographically Weighted Regression analysis combined with Bayesian Maximum Entropy theory (BME-GWR) (Xiao et al., 2018), space-time random forest (STRF) model (Wei et al., 2019), and space-time support vector regression (STSVR) model (Yang et al., 2018), which can not only consider the spatiotemporal heterogeneity of snow density, but also achieve snow density prediction without ground observation. Of course, if the collected data and the scale are allowed to run a physical model, it would be better to combine the physical model and the machine learning models. Hope our efforts have addressed your concern.

---

## Author Response (AR2)

**Response to editors**

Tc-2022-45 "Towards Large-Scale Daily Snow Density Mapping with Spatiotemporally Aware Model and Multi-Source Data"

The authors greatly appreciate for your constructive comments and kind suggestions. We have corrected the manuscript accordingly. Below we will address each comment in a point-by-point answer:

- 5
- Bold: comments of the editor
- blue words: answer of the authors
- Italics or red words: changes to the initial manuscript

Thank you once again for your help to our paper.

10

**Comments by Dr. Chris Derksen**

**1. Line 20: change to "20 potentially influencing variables."**

We have modified in the manuscript.

**15 2. Line 21: change to "an R2 of 0.531"**

We have modified in the manuscript.

3. Line 24: change to "...the GWTNN model in capturing..."

We have modified in the manuscript.

4. Line 57: "For example, snow density was found to be lower at higher elevations, and even decreased by approximately 0.006 g/cm3 with each 100 m increase in elevation (Zhong et al., 2014)" Can you add specification to the geographic region for which this result from the Zhong et al study applies?

We have added the geographic region of this result in the manuscript.

5. Line 61: remove "In addition, the aspect also affects the snow density through radiation, because sunnyfacing slopes that experience" and just start the sentence with "Slopes with high radiation input will be more likely..."

**25 likely.**

We have modified in the manuscript.

6. Line 99: change 'various' to 'variable'

We have modified in the manuscript.

**7. Line 101: change to "satellite data can provide information on the snow-related..."**

30 We have modified in the manuscript.

**8. Figure 1: remove the inset map. It is not needed given the geographic focus of the analysis.**

Thank you for the suggestion. We truly agree with this. Considering that snow is mainly distributed in high altitude and high latitude areas, we made the small inset map in southern China to highlight the snow areas in Figure 1.

**9. Line 145: change to "...different influencing variables on snow density."**

35 We have modified in the manuscript.

10. Line 151: consider adding more detail to this statement: "In addition, the min-max normalization method is applied to normalize different influencing variables." Can you point to which variables are normalized in Figure 2?

Thanks for this comment. we have added the aim of normalization, and explained that all the influencing variables will be normalized except for MCD12Q1 data, as below.

**2.3 Data Integration**

In addition, to eliminate the influence of different dimensions, the min-max normalization method is applied to normalize different influencing variables except for MCD12Q1 data.

**11. Line 232: change 'involved' to 'included'**

**45 We have modified in the manuscript.**

**12. Line 268: change to "...caused by greater forest cover..."**

We have modified in the manuscript.

**13. Line 346: change to "...also provides gridded daily snow density data..."**

We have modified in the manuscript.

50 14. Line 398-399: remove "in different periods and regions."

We have modified in the manuscript.

15. Line 469: change to "...based on the observed snow density."

We have modified in the manuscript.

**16. Line 480: I suggest adding an extra statement to the new text which emphasizes that the individual correlations with the 20 variables were generally weak, as you note on line 238.**

We added an extra statement according in the manuscript, as below.

**6** Conclusion**

The individual correlations between snow density and 20 influencing variables are very week, with the maximum average correlation coefficient of only 0.123, and it is found that the vegetation variable LAI\_HV, meteorological variable TP, snow variable SD, and topographic variables have a relatively close relationship to snow density.

60 var

Thank you very much.

**Comments by Polina Shvedko**

**65 The table is included as figure (Figure 4). Please re-label this as table and the references in the manuscript text must be adjusted accordingly. If the color spectrum of these tables is necessary and cannot be exchanged for footnotes, bold, or italic, then the table must be inserted as an image, but still be called a table.**

70

Thank you very much for this comment. Actually, the Figure 4a is a heat map with label rather than a table. the numbers in this figure are labeled to better illustrate the individual correlations in addition to the color spectrum. Hence, we suggest that the figure would be a better choice to present the result. In addition, the data in Figure 4b comes from Figure 4a, and Figure 4a and Figure 4b work together to reveal the correlations between snow density and its influencing variables. If we changed Figure 4a as a table, then Figure 4b is needed to be a separate figure. Considering the close relation between Figure 4a and Figure 4b, we suggest not to separate them. According to the above considerations, we suggest to keep Figure 4 as is. Hope our explanations could address your major concern. Thanks again.